# Impact of in-cloud aqueous processes on the chemical compositions and morphology of individual atmospheric aerosols

Yuzhen Fu[1, 2], Qinhao Lin[1, #], Guohua Zhang[1, 3, *], Yuxiang Yang[1, 2], Yiping Yang[2, 4], Xiufeng Lian[1, 2], Long Peng[1, 2], Feng Jiang[1, 2, ##], Xinhui Bi[1, 3, *], Lei Li[5], Yuanyuan Wang[6], Duohong Chen[7], Jie Ou[8], Xinming Wang[1, 3], Ping'an Peng[1, 3], Jianxi Zhu[4], Guoying Sheng[1]

[1] State Key Laboratory of Organic Geochemistry and Guangdong Key Laboratory of Environmental Protection and Resources Utilization, Guangzhou Institute of Geochemistry, Chinese Academy of Sciences, Guangzhou 510640, PR China

[2] University of Chinese Academy of Sciences, Beijing 100049, PR China

[3] Guangdong-Hong Kong-Macao Joint Laboratory for Environmental Pollution and Control, Guangzhou 510640, PR China

[4] CAS Key Laboratory of Mineralogy and Metallogeny & Guangdong Provincial Key Laboratory of Mineral Physics and Materials, Guangzhou Institute of Geochemistry, CAS, Guangzhou 510640, PR China

[5] Institute of Mass Spectrometer and Atmosphere Environment, Jinan University, Guangzhou 510632, PR China

[6] Department of Atmospheric Science, School of Earth Science, Zhejiang University, Hangzhou 310027, PR China

[7] State Environmental Protection Key Laboratory of Regional Air Quality Monitoring, Guangdong Environmental Monitoring Center, Guangzhou 510308, PR China

[8] Shaoguan Environmental Monitoring Center, Shaoguan 512026, PR China

[#] now at: Guangdong Key Laboratory of Environmental Catalysis and Health Risk Control, Guangzhou Key Laboratory Environmental Catalysis and Pollution Control, School of Environmental Science and Engineering, Institute of Environmental Health and Pollution Control, Guangdong University of Technology, Guangzhou 510006, PR China.

[##] now at: Institute of Meteorology and Climate Research, Karlsruhe Institute of Technology, Eggenstein-Leopoldshafen 76344, Germany.

*Correspondence to*: Guohua Zhang (zhanggh@gig.ac.cn) and Xinhui Bi (bixh@gig.ac.cn)

**Abstract.** The composition, morphology, and mixing structure of individual cloud residues (RES) and interstitial particles (INT) at a mountain-top site were investigated. Eight types of particles were identified, including sulfate-rich (S-rich), S-organic matter (OM), aged soot, aged mineral dust, aged fly ash, aged metal, refractory, and aged refractory mixture. A shift of dominant particle types from S-rich (29%) and aged soot (27%) in the INT to S-OM (24%) and aged refractory mixture (22%) in the RES is observed. In particular, particles with organic shells are enriched in the RES (30%) relative to the INT (12%). Our results highlight that the formation of more oxidized organic matter in the cloud contributes to the existence of organic shells after cloud processing. Fractal dimension ($D_f$), a morphologic parameter to represent the branching degree of particles, for soot particles in the RES ($1.82 \pm 0.12$) is lower than that in the INT ($2.11 \pm 0.09$), which indicates that in-cloud processes may result in less compact soot. This research emphasizes the role of in-cloud processes on the chemistry and microphysical properties of individual particles. Given that organic coatings may determine the particle hygroscopicity, activation ability, and heterogeneous chemical reactivity, the increase of OM-shelled particles upon in-cloud processes should have considerable implications.

## 1 Introduction

Aerosol-cloud interaction is regarded as one of the most significant sources of uncertainty in assessing the radiative forcing of aerosols so far (IPCC, 2013). On the one hand, aerosols can participate in the formation of cloud droplets, which is primarily influenced by their chemical composition and size (Fan et al., 2016; Maskey et al., 2017; Ogawa et al., 2016; Raymond and Pandis, 2002; Zelenyuk et al., 2010). On the other hand, in-cloud processes, including the formation of sulfate, nitrate, and water-soluble organics, and the physical processes such as collision and coalescence, would substantially change the physical and chemical properties of the activated particles (Kim et al., 2019; Ma et al., 2013; Roth et al., 2016; Wu et al., 2013). Given that the morphology and mixing state are vital in determining the optical properties of particles (Adachi et al., 2010; Wu et al., 2018), changes of these properties upon in-cloud processes would further affect the subsequent atmospheric processes (e.g., cloud activation, heterogeneous reactions) and radiative forcing of particles after droplet evaporation.

Understanding the morphology and mixing state of particles upon in-cloud processes is of considerable significance to improve the knowledge of aerosol-cloud interactions. For instance, Zelenyuk et al. (2010) found that both cloud droplet residues (RES) and interstitial particles (INT, or unactivated particles in the cloud) are mainly composed of organics, sulfate, biomass burning particles, and processed sea salt at the North Slope of Alaska. Kamphus et al. (2010) observed that 92% of RES are particles containing sulfates, organics, and nitrate at the Jungfraujoch (Swiss Alps). At Mt. Tai, Liu et al. (2018b) observed that the main particle types are S (sulfate)-soot (36%), S-fly ash/metal-soot (26%), and S-rich (24%) for RES and S-rich (61%), S-soot (15%) and soot (15%) for INT. These results indicate that both RES and INT present complex mixtures, and carbonaceous matters (i.e., organic materials (OM) and soot) are critical materials in the cloud mass.

While extensive studies are reporting the extent of aqueous phase processing on the modification of aerosol bulk (e.g., mass) and/or chemical (e.g., mixing state, hygroscopicity) properties (Chakraborty et al., 2016; Ervens et al., 2011), the influence of in-cloud processes on the physical properties (e.g., shape, mixing structure) of individual particles is still ambiguous. In particular, physical properties play a leading role in the cloud activation of inorganic/organic mixed particles (Topping et al., 2007). A hydrophobic organic-rich coating will form on a hygroscopic particle core if liquid-liquid phase separation occurs (Song et al., 2013). Besides, the distribution of organics and its association with other aerosol types is also crucial for the accurate calculation of its radiative effects (Zhu et al., 2017). However, to what extent in-cloud processes play a role in reshaping the distribution of organic and inorganic compositions remains unknown, although such coating structures have been identified in ambient aerosols (Adachi and Buseck, 2008; Li and Shao, 2010; Yu et al., 2019). Considering that secondary formation during in-cloud processes contributes to a substantial fraction (up to 60%) of organic aerosols (Ervens et al., 2011; Liu et al., 2012; Myriokefalitakis et al., 2011; Spracklen et al., 2011), the influence of this process in atmospheric chemistry cannot be neglected.

For another type of carbonaceous material (i.e., soot), there is extensive evidence showing that the absorption and cloud activation of soot-containing particles can be significantly affected by coatings (Adachi et al., 2010; Wu et al., 2018; Moffet and Prather, 2009). The critical factors to accurately predict such impact include the

amount and nature of the coating material, the exact particle morphology, and the size distribution (Qiu et al.,
2012; Radney et al., 2014). Fractal dimension ($D_f$) is widely used to indicate the extent of branching of soot (Brasil
et al., 1999), with densely packed or compacted soot particles having higher $D_f$ than chain-like branched clusters
or open structures. When the branched soot particles become compact, their size will decrease, but the scattering
cross-section will be greater (Radney et al., 2014; Zhang and Mao, 2020). While some studies have found that
soot restructuring occurs after aqueous processing (Bhandari et al., 2019; Ma et al., 2013; Mikhailov et al., 2006),
or being coated by OM (Spencer and Prather, 2006) and sulfate (Zhang et al., 2008), Khalizov et al. (2013)
suggested that soot with thin organic coating did not become more compact under high humidity. Besides, the
morphology and mixing structure of soot involving the formation of organics upon cloud processing is also poorly
constrained.
To further improve our understanding of the morphology and mixing structures between the various
components within individual RES and INT, we conducted a 25-day field observation of cloud events at a
background site in southern China. A transmission electron microscope (TEM) combined with energy-dispersive
X-ray spectrometry (EDS) was used to analyze the chemical composition, size, morphology, and mixing structure
of individual RES and INT. Previously, the chemical composition and mixing state of RES at the same site have
been investigated with a single particle aerosol mass spectrometer (SPAMS) (Lin et al., 2017; Zhang et al., 2017a).
Herein, we focus on the mixing structure (e.g., chemical compositions and morphology) of individual particles,
in particular, OM-containing particles. Meanwhile, particle types and mixing state of RES and INT are also
discussed. The difference between the mixing structure of RES and INT may indicate the impact of in-cloud
aqueous processes.

## 2 Materials and Methods

### 2.1 Sampling site

Sampling was conducted at the top of Mt. Tianjing (112°53′56″ E, 24°41′56″ N; 1690 m above sea level) in
southern China from 18 May to 11 June 2017. The sampling site is located in a natural preserve, and it is almost
unaffected by local anthropogenic sources. It is about 50 km and 350 km away from the north of the Pearl River
Delta (PRD) region and the South China Sea, respectively.

### 2.2 Collection of RES and INT

A cloud event was identified with visibility below a threshold of 3 km and relative humidity (RH) above a
threshold of 95%, using a ground-based counterflow virtual impactor (GCVI, model 1205, Brechtel Mfg. Inc.,
USA). The GCVI was automatically triggered when there was a cloud event, whereas it was not allowed to sample
when a precipitation sensor detected rain or snow. Then cloud droplets were introduced into the GCVI, followed
by removing water in an evaporation chamber (40 ℃) to obtain RES. The sampling process might experience
some particle loss due to the evaporation of highly volatile substances. The droplet cut size, at which the
transmission efficiency of CVI is 50%, was set at a size larger than 7.5 μm (Shingler et al., 2012). INT was
sampled using another inlet (PM$_{2.5}$ cyclone inlet, with a flowrate of 5 lpm), followed by passing through a silica
gel diffusion dryer.
A DKL-2 sampler (Genstar Electronic Technology Co., Ltd., China) was used to collect RES and INT on copper
grids coated with carbon film with an airflow of 1 L min$^{-1}$. The collection efficiency of the sampler is 50% at a
particle size of 80 nm, assuming the particle density is 2 g cm$^{-3}$. To avoid particle overlapping, the sampling
duration was set within 10 minutes. All samples were placed in a sealed plastic sample box and stored in a
desiccator at room temperature for subsequent analysis.
The information about cloud events and samples are summarized in Table 1. We focused on three cloud events
(#1, #2, and #3), with a duration of 14, 34, and 47 hours, respectively. RES and INT samples from these cloud
events were analyzed, with INT not available for the cloud event #1. To minimize the influence of rapid change
of cloud condition, all the samples were collected during the stable and mature periods (Visibility < 100 m).

## 2.3 TEM analysis of RES and INT

Chemical composition, size, and morphology of individual RES and INT were characterized by a TEM (FEI
Talos F200S) operated at 200 kV. TEM/EDS is a very effective tool to analyze the microscopic characteristics of
individual particles. The resolution of images between 1 μm and 100 nm can be magnified from 7,000 to 36,000
fold, which depended on the size of particles. The EDS is coupled with TEM to detect the intensity of elements
including carbon and heavier elements ($Z \geq 6$). The produced X-rays signal in the EDS system is detected by a
silicon (Si) drift detector (SDD), and thus Si is not considered in the discussion. Cu is also not considered due to
the interference from the copper grids. In the TEM vacuum chamber, some volatile substances (e.g., ammonium
nitrate (NH$_4$NO$_3$) and volatile organic matter) would be lost. Moreover, volatile materials are often sensitive to
strong electron beams. Due to the analysis error of volatile materials, TEM/EDS studies typically focus on
refractory compositions. Using an image analysis software (ImageJ), the equivalent circle diameters (ECD) of all
particles can be obtained from the scanned images from the TEM. For particles with rim, only the nucleus is
counted, because the rims contain only a small amount of OM. Overall, 780 particles, including RES and INT,
were analyzed.
Base on various element spectra, RES and INT were mainly classified as sulfate-rich (S-rich), carbonaceous
material, mineral dust, metal, and fly ash (Li et al., 2016; Twohy and Anderson, 2008). Elemental compositions
of S-rich particles were dominated by S and O, and some of them were associated with minor N, K and Na. Low
intensity of N could be due to the evaporation of ammonia nitrate under the high energy electron beam (Smith et
al., 2012). This led to the bubbly appearance of S-rich particles. In this case, S-rich particles represented secondary
inorganic particles. The elemental compositions of carbonaceous materials were characteristics of abundant C and
minor O. Carbonaceous materials were divided into soot and OM according to different morphology. Soot was
composed of tens to hundreds of carbon spheres ranging from 21 to 108 nm in diameter (average diameter was
47.7 nm), which often displayed botryoidal aggregates. OM did not have a chain-like structure, which generally
exhibited amorphous state and spherical or irregular shapes. Mineral dust particles were consisted of Si, Al, Ca,
O and minor Fe. Mineral dust was mainly clay, feldspar, calcite and gypsum, usually showing irregular shapes.
Metal particles were represented as Fe, Zn, Ti, Mn, or Ni. Metal particles were characteristic of spherical,

rectangular or irregular morphologies. They were largely from natural dust and industrial combustion (Moffet et al., 2008; Silva et al., 2000; Ye et al., 2018). The presence of spherical metal particles indicated that they experienced melting at high temperature (Giere et al., 2003; Giere et al., 2006). Fly ash particles mainly contained Si, Al and O. Fly ash particles tended to be spherical in morphology and they were generally produced from the process of coal combustion (Chen et al., 2012; Henry and Knapp, 1980).

**2.4 SPAMS analysis of RES and INT**

A SPAMS (Hexin Analytical Instrument Co., Ltd., Guangzhou, China) was used to analyze the chemical composition and size distribution of individual particles in real-time. Particles entering the SPAMS were first focused into a beam of particles through an aerodynamic lens, and then their flight velocities were determined by two continuous diode Nd:YAG laser beams (532 nm). Polystyrene spheres of known size were used as a standard substance to calibrate the vacuum dynamic size ($d_{va}$) of particles. Next, the pulsed laser (266 nm) was precisely triggered to ionize the target particle according to the intrinsic velocity of each particle, and the positive and negative ions are separated and analyzed using a dual polarity time-of-flight mass analyzer. Finally, we obtained the information of individual particles, including $d_{va}$ and the positive and negative ion mass spectra. The relative peak area of characteristic peaks for each species in the mass spectra is generally applied to indicate its relative abundance in the particle (Bhave et al., 2002; Gross et al., 2000). However, it is still challenging to provide quantitative information on chemical compositions, mainly attributed to the different ionization efficiency and the complex matrix effects for various types of particles. A detailed description of particle analysis methods and particle type characteristics can be found in the supporting information.

**2.5 Calculating morphology parameters of soot**

The fractal dimension of soot is characterized in the following statistical scaling law (Brasil et al., 1999; Köylü et al., 1995):

$$N = k_g \left( \frac{2R_g}{d_p} \right)^{D_f}$$

where $N$ is the number of monomers within a certain soot aggregate, $k_g$ is the fractal pre-factor, $R_g$ is the radius of gyration, $d_p$ is the diameter of the monomer, and $D_f$ is the mass fractal dimension. $R_g$ can be obtained by using a simple relationship between $R_g$ and $L_{max}$, the maximum length of the soot aggregate (Brasil et al., 1999):

$$L_{max}/2R_g = 1.50 \pm 0.05$$

And, the number of monomers, $N$, can be calculated by a power-law correlation of projected area of monomer and aggregate:

$$N = k_a \left( \frac{A_a}{A_p} \right)^{\alpha}$$

where $k_a$ is a constant, $A_a$ and $A_p$ are the projected area of aggregate and monomer, respectively, and $\alpha$ is an
empirical projected area exponent. The value of $k_a$ and $\alpha$ depends on the degree of monomer overlap ($\delta$) in the
aggregate (Oh and Sorensen, 1997), and $\delta$ can be determined by:

$$\delta = \frac{2a}{l}$$

where $a$ is monomer radius, and $l$ is the center distance of adjacent monomers. The values of parameters including
$a$, $l$, $A_a$, $A_p$, $L_{max}$, and $d_p$ can be obtained by analyzing TEM images. Then $D_f$ can be calculated by the above four
formulas.

## 3 Results and Discussion

### 3.1 Particle type and mixing state of RES and INT

According to mixing state, RES and INT were divided into the following eight types (Figure 1): S-rich, S-OM,
refractory (soot/mineral dust/metal/fly ash), aged soot (S/OM-soot), aged mineral dust (S/OM-mineral dust), aged
metal (S/OM-metal), aged fly ash (S/OM-fly ash), and aged refractory mixture (S/OM-soot/mineral dust/metal/fly
ash). S-rich or OM, generally considered to be aged since they are mainly secondarily produced in the atmosphere,
are internally mixed with refractory materials (soot/mineral dust/metal/fly ash) (Canagaratna et al., 2007; Huang
et al., 2012; Jiang et al., 2019). Such internally mixed S/OM-refractory particles are named as aged refractory
particles herein. Aged particle types containing two or more refractory components are named as "aged refractory
mixture". It is worth noting that refractory are refractory particles without S-rich and OM.
Figure 2 shows the number fraction of different particle types in the RES and INT during cloud events #2 and
#3. S-rich, S-OM, aged soot, and aged refractory mixture particles are dominant particle types. The most abundant
particles in the RES are aged refractory mixture (23%), followed by S-OM (22%), aged soot (20%), S-rich (16%),
aged metal (9%), aged fly ash (5%), aged mineral dust (4%), and refractory (1%). Differently, INT is
predominated by S-rich (29%), aged soot (27%), S-OM (15%), aged refractory mixture (10%), and the lesser
percentage of aged fly ash (8%), refractory (5%), aged mineral dust (4%), and aged metal (2%) were also observed.
Among three cloud events, the RES are dominated by S-OM in cloud event #1 and #2 and aged refractory mixture
particles in cloud event #3 (Figure 3). It is also shown that the RES and INT analyzed by TEM/EDS can represent
their compositions throughout cloud events #2 and #3, since such compositions were relatively stable throughout
these periods (Figure S3).
The different air masses are expected to affect the distribution of particle types. The distribution of several types
of particles in the RES was observed to be divergent in different cloud events, corresponding to different air
masses, as shown in Figure 3 and Figure 4. The number fraction of OM-containing particles was the highest (81%)
in cloud event #2, which might be partly attributed to the higher concentration of $O_3$ during cloud event #2 (Table
S1). And the samples of cloud event #2 were collected at noon. Higher solar radiation during the sampling time
might also promote heterogeneous photochemical oxidation reactions during the cloud process and increased the
generation of OM within cloud droplets (Xu et al., 2017). Aged metal particles accounted for a similar percentage
(7-12%) for three cloud events. The proportion of aged mineral dust during cloud event #1 (14%) was nearly four
times those in the other two cloud events. Aged fly ash particles had the highest proportion (10%) in cloud event
#3 compared with the other two cloud events, most probably influenced by the different air masses (Figure 4).
Aged mineral dust particles of cloud event #1 may be influenced by the long-distance transportation of dust from
Southeast Asia (Salam et al., 2003). Clearly, aged fly ash particles of cloud event #3 are associated with the air
masses from the PRD region with a dense distribution of industrial facilities there (Cao et al., 2006).
**3.2 The morphology and mixing structure of carbonaceous particles**
OM-containing particles, including all of S-OM particles, part of aged refractory (S-OM/OM-refractory) and
aged refractory mixture (S-OM/OM-soot/mineral dust/metal/fly ash) particles, accounted for 60% of RES and 33%
of INT during cloud events #2 and #3. According to the mixing structures between OM and other materials (Figure
5), OM-containing particles are classified into the following five categories: thinly coated (Figure 5b), core-shell
(Figure 5c), embedded (Figure 5d), attached (Figure 5e), and homogenous-like (Figure 5f) structures (Li et al.,
2016). A particle is classified as a thinly coated structure when wrapped with a thin layer of OM. The thickness
of the OM layer of thinly coated particles ranges from 12 to 150 nm. Generally, the shapes of OM-containing
particles with the thinly coated structure are elliptical or irregular. The difference between the core-shell structure
and thinly coated structure is the relative thickness of OM: Core-shell structure possessed thicker organics than
thinly coated structure. The thickness of the shell varies from 86 to 2110 nm, and the ratio of the projected area
of the shell to particle ranges from 0.20 to 0.97. Moreover, OM-containing particles with a core-shell structure
are round. Embedded or attached structure for the OM-containing particles refers to the relative distribution of
OM, i.e., embedded in or attached to other materials (e.g., sulfate). Well mixed OM-containing particles with no
identifiable boundary between organic and non-organic matter were identified with a homogenous-like structure.
The first most abundant particles are thinly coated geometry, comprising 53% of RES and 59% of INT during
cloud event #2 and #3. The second are core-shell particles for RES and attached particles for INT. The percentage
of core-shell particles in the RES is almost 2.5 times that in the INT (27% vs. 12%). Embedded and homogenous-
like particles account for minor proportions (< 4%) for both RES and INT.
Soot-containing particles, including all of the aged soot particles (S/OM-soot) and part of refractory
(soot/mineral dust/metal/fly ash) and aged refractory mixture particles (S/OM-soot/mineral dust/metal/fly ash),
account for 36% of RES and 39% of INT during cloud event #2 and #3, respectively. The fraction is consistent
with the range of those (< 30% – ~60%) observed at the same site by SPAMS (Zhang et al., 2017a). Most of the
soot particles are observed to distribute around the periphery of particles (Figure S4).
**3.3 In-cloud formation of OM**
It can be seen from Figure 2 that a shift of dominant particle types from S-rich (29%) and aged soot (27%) in
the INT to the aged refractory mixture (23%) and S-OM (22%) in the RES. In particular, the fraction of OM-
containing particles increases from 33% in the INT to 60% in the RES. It is unlikely due to the favorable activation
of S-OM or aged refractory mixture, since mixing with OM generally lower the hygroscopicity of inorganic-
dominant particles (e.g., S-rich) (Brooks et al., 2004; Pierce et al., 2012). OM coating at the same site has been

shown to inhibit the CCN activation of soot-containing particles (Zhang et al., 2017a). Instead, it is most probably attributed to the in-cloud formation of OM on the surface of some S-rich particles, shifting the dominant particle type from S-rich to S-OM particles. It can be supported by the relatively larger median size of S-OM particles (0.76 μm) than S-rich particles (0.56 μm) (Figure S5), since in-cloud formation of OM is expected to enlarge the original S-rich particles (Pierce et al., 2012).

In addition, the fraction of OM-containing particles with core-shell mixing structure in the RES is almost 2.5 times that in the INT (Figure 5a). Such a mixing structure is similar to those observed in the Arctic, background, or rural atmosphere (Hiranuma et al., 2013; Li et al., 2016; Yu et al., 2019), but is different from other findings in polluted areas, where OM-containing particles mainly existed in homogenous-like and thinly coated structures (Li et al., 2016). It is also consistent with several laboratory simulations demonstrating that reactive uptake of volatile organic compounds (VOCs) on inorganic sulfate and heterogeneous and multiphase reactions between there species would lead to a core-shell morphology (e.g., Riva et al., 2019; Zhang et al., 2018a; Zhang et al., 2019). Recently, Gorkowski et al. (2020) came up with a particle morphology prediction framework developed for mixtures of organic aerosol based on the measurements from aerosol optical tweezers experiments and literature data, and they hypothesized the core-shell morphology dominated by secondary organic aerosols (SOA) in the shell phase.

Moreover, we estimated the O/C ratio of coating and shell within OM-containing particles. It should be noted that the O/C ratio of organic coating and shell is underestimated herein due to the copper grid evenly covered by carbon film. Moreover, while some loss of volatile organic compounds during the TEM/EDS analysis may affect the O/C of particles, the relatively higher O/C ratio for the RES is still affirmative. Droplets are expected to dissolve more volatile organic compounds (Chakraborty et al., 2016) with higher O/C. The release of these compounds during droplet evaporation would result in underestimating of O/C in the RES. We found that the average value of the O/C ratio of RES is higher than that of INT, and the average value of the O/C ratio of RES with a core-shell structure is 0.23, which is two times that with a thinly coated structure (0.11) (Table 2), indicating that these RES with core-shell particles are more oxidized. At the same site, we have previously observed enhanced aqueous SOA products, such as oxalate in the cloud (Zhang et al., 2017b). The higher O/C ratio of core-shell particles is also consistent with current studies reporting more oxidized organic species in cloud/fog residues (Brege et al., 2018; Chakraborty et al., 2016; Zhang et al., 2017b). With high levels of VOCs at the sampling site (Lv et al., 2019), the prevalent formation of aqueous SOA through the uptake of VOCs in cloud droplets would be expected (Kim et al., 2019; Liu et al., 2018a). The contribution from photochemical processes may also be reflected by the association of the highest fraction (81%) of OM-containing particles with a higher concentration of $O_3$ during cloud event #2 (Table S1). Consistently, the relative peak area of $m/z$ 43$C_2H_3O^+$ in the RES is higher than that in the INT during cloud event #2 (Figure S7), indicative of the favorable formation of oxidized organic compounds (Qin et al., 2012; Zhang et al., 2017b).

However, one may expect that such a core-shell mixing structure in the RES can also be explained by the primary activation of S-OM particles with larger sizes. Unfortunately, no sample before the cloud events is available for TEM/EDS measurements. However, with evidence from the collocated SPAMS, we show that this is not convincing. As shown in Table S2, the ratios of relative peak area between organics and sulfate are similar

between the INT and particles before cloud event, whereas they are higher in the RES. This is corresponding to
the production of oxidized organics during in-cloud processes (Zhang et al., 2017b), consistent with the TEM/EDS
results.
**3.4 The $D_f$ of soot in the RES and INT**
Figure 6 shows the $D_f$ of soot within RES and INT of cloud event #2 and #3. The result shows that the $D_f$ of
soot is smaller in the RES (1.82 ± 0.12) than in the INT (2.11 ± 0.09), which means that soot is more branched in
the RES. It is noted that 62.5% of all soot-containing particles with clear boundaries are included in the $D_f$
calculation since thick coating around soot might make the boundary of monomers not clear enough (Bhandari et
al., 2019). The obtained $D_f$ are close to those (1.83 – 2.16) reported at a background site (Wang et al., 2017). The
$D_f$ of soot in the RES and INT likely represents partly coated soot (1.82 ± 0.05) (Yuan et al., 2019) and embedded
soot (2.16 ± 0.05) (Wang et al., 2017), respectively. In addition to emission sources and coating processes, high
relative humidity (RH) during nighttime is a critical factor to increase the compactness of soot (Yuan et al., 2019).
While some previous studies demonstrated that soot aggregates tend to be more compact (with larger $D_f$) after
aging or cloud processing (Adachi and Buseck, 2013; Moffet and Prather, 2009; Wu et al., 2018), our results
suggest that in-cloud processes may result in more branched soot, as shown in Figure 6. Considering that $D_f$ is
controlled mainly by emission sources, combustion conditions, and aging processes (Adachi et al., 2007), we
propose three possible explanations for the lower $D_f$ of soot in the RES than that in the INT. The first and the most
likely reason is that some of the soot aggregates are immediately encapsulated by non-volatile materials (such as
organic matter) after emission by combustion sources. These coatings fill the spaces between the branches of soot
aggregates, which inhibits the relatively large deformation and reconfiguration of the soot aggregates during
transport and activation into cloud droplets (Zhang et al., 2018b). Differently, soot aggregates may shrink easily
and become more compact during the long-distance transport if the soot aggregates are emitted without non-
volatile coatings (Adachi and Buseck, 2013). We show that soot aggregates have higher $D_f$ and lower average
ECD in the INT (247 nm) than in the RES (266 nm), which means that larger, less dense soot particles are easier
to act as CCN. This is consistent with a study reporting that small particles are more compact than large particles
(Adachi et al., 2014). The second is that water-soluble substances within aerosols will be miscible after activating
to cloud droplets (Gorkowski et al., 2020). The coating materials of soot may be released, which makes soot more
branched in the droplets and the following-up droplet evaporation. The third possible explanation is that different
combustion materials and combustion conditions produce soot-containing particles with different mixing states
and morphology (China et al., 2014; Khalizov et al., 2013; Liu et al., 2017; Zhang et al., 2018b).
This result contrasts with the current study reporting that soot sampled after cloud droplet evaporating is more
compact than freshly emitted and interstitial soot (Bhandari et al., 2019). Our observations at the background site
show that the majority of soot aggregates in both RES and INT (~80%) are located in off-center positions (Figure
S4), having less compact shapes even after being coated. This is quite different from the core-shell model currently
used in the climate models (Bond and Bergstrom, 2006; Wu et al., 2018). Through theoretical calculation, Adachi
et al. (2010) suggested that absorption cross-sections could be reduced by 20-30% with off-center positions of
soot relative to center positions. This means that the models based on core-shell assumption may overestimate the
absorption of soot-containing particles after cloud processing.

**4 Conclusion and atmospheric implications**

The result highlights the different morphology and mixing structures of activated and interstitial particles, which may imply the substantial role of in-cloud aqueous processes in reshaping the activated particles. While Yu et al. (2019) considered organic coatings on sulfate in the Arctic as a result of the increase of SOA following particle aging and growth during transport, our data further imply a specific role of in-cloud processes in the coating on sulfate. The prevalence of OM shelled particles after cloud processing also supports a current laboratory observation depicting that rapid film formation and fast heterogeneous oxidation can provide an efficient way of converting water-insoluble organic films into more water-soluble components in aerosols or cloud droplets (Aumann and Tabazadeh, 2008).

Gorkowski et al. (2020) suggested that mixing structures of OM-containing particles is related to the oxidation degree of OM. We also show that OM shells formed in cloud droplets have a higher degree of oxidation. Such a chemical and morphological modification of aerosol particles may influence species diffusivities from the interior to the surface region of the shell and gas-particle partitioning between the shell and gas (Liu et al., 2016; Shiraiwa et al., 2013). Such a reshaping may also have an influence on aerosol hygroscopicity. Extrapolating the linear relationship between the O/C ratio and the hygroscopicity parameter ($\kappa_{org}$) indicates that $\kappa_{org\text{-}shell}$ is about 1.4 times $\kappa_{org\text{-}coating}$ (Jimenez et al., 2009; Lambe et al., 2011). In addition, the formation of the organic film could result in a change of surface tension and thus affect the critical supersaturation required for particle activation (Ovadnevaite et al., 2017). The heterogeneous ice nucleation potential may be suppressed for mineral particles when coated by OM (Möhler et al., 2008). Given the critical contribution of in-cloud aqueous SOA, several mixing structures of OM-containing aerosols upon in-cloud processes may have substantial implications in modeling the direct and indirect radiative forcing of aerosols (Scott et al., 2014; Zhu et al., 2017).

*Data availability.* Data are available on request from Guohua Zhang (zhanggh@gig.ac.cn) and Xinhui Bi (bixh@gig.ac.cn).

*Author contribution.* GHZ and XHB designed the research (with input from XMW and GYS). YZF, GHZ, and XHB analyzed the data, and wrote the manuscript. YZF, XFL, YXY, FJ, and QHL conducted sampling work under the guidance of GHZ, XHB and XMW. LL, DHC and JO had an active role in supporting the sampling work. YZF performed the laboratory analysis of individual particles by TEM/EDS, with support from YPY and JXZ. All authors contributed to the discussions of the results and refinement of the manuscript.

*Competing interests.* The authors declare that they have no conflict of interest.

*Acknowledgements.* This work was supported by the National Nature Science Foundation of China (No. 41775124 and 41877307), Natural Science Foundation of Guangdong Province (2019B151502022), and Guangdong Foundation for Program of Science and Technology Research (Grant No. 2019B121205006 and 2017B030314057). The authors gratefully acknowledge the NOAA Air Resources Laboratory (ARL) for the provision of the HYSPLIT transport and dispersion model (http://ready.arl.noaa.gov) used in this publication.

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

**Table 1. The information of cloud events and samples, including starting and ending time of each cloud event, the number and type of analyzed particles, the mean value**
**of visibility and number concentration of RES or INT during sampling time.**

| Cloud event | Starting Time* | Ending Time* | Particles | Type | Visibility/m | Number Concentration/cm$^{-3}$ |
|---|---|---|---|---|---|---|
| Cloud #1 | 2017/5/20 18:19 | 2017/5/21 8:34 | 190 | RES | 66 | 195 |
| Cloud #2 | 2017/5/23 20:35 | 2017/5/25 6:35 | 161 | INT | 50 | 99 |
| | | | 162 | RES | 88 | 299 |
| Cloud #3 | 2017/6/8 18:30 | 2017/6/10 17:30 | 132 | INT | 44 | 996 |
| | | | 135 | RES | 33 | 111 |

* Local time, i.e., Chinese Standard Time, UTC+8.
**Table 2. The average value of O/C ratio of OM-containing particles with thinly coated and core-shell mixing structures.**

| Type | thinly coated | core-shell |
|------|---------------|------------|
| RES | 0.11 | 0.23 |
| INT | 0.08 | 0.06 |


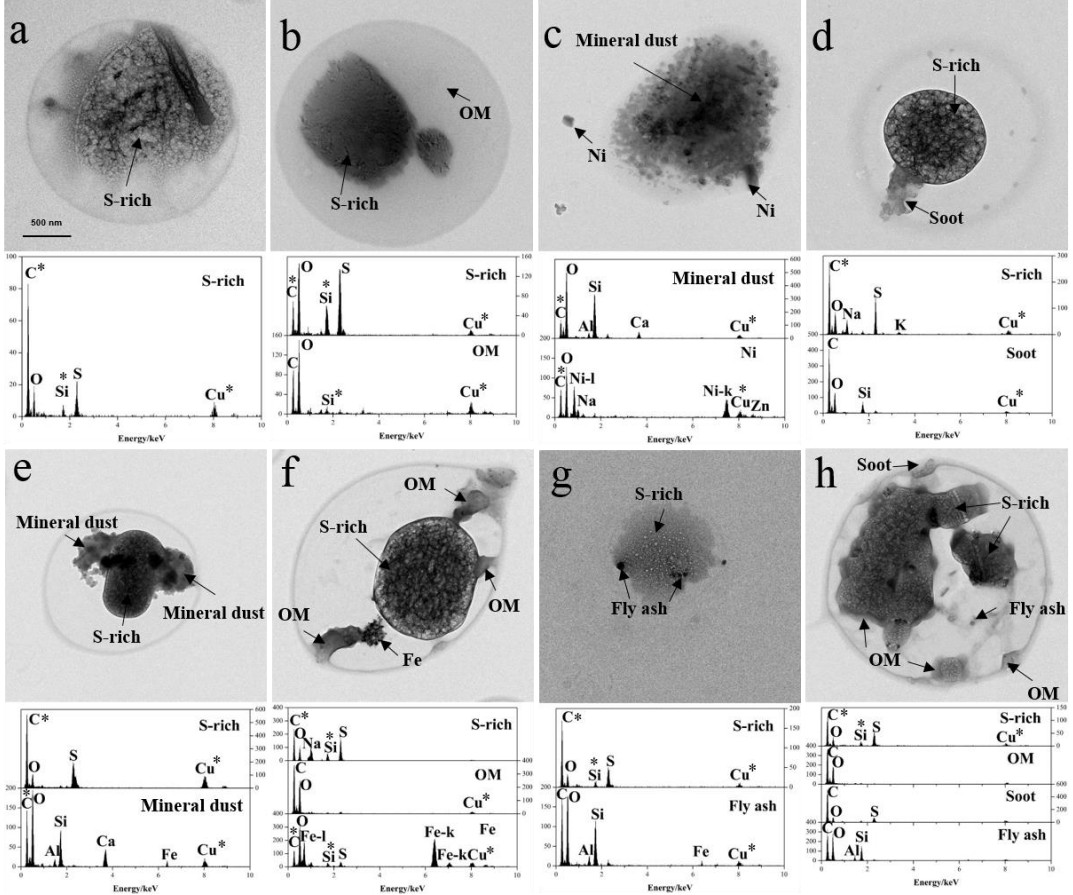


**Figure 1. TEM images and EDS spectra of individual RES and INT particles with different particle types: (a) S-rich; (b) S-OM; (c) refractory; (d) aged soot; (e) aged mineral dust; (f) aged metal; (g) aged fly ash; (h) aged refractory mixture. Asterisk (*) represents the background element.**

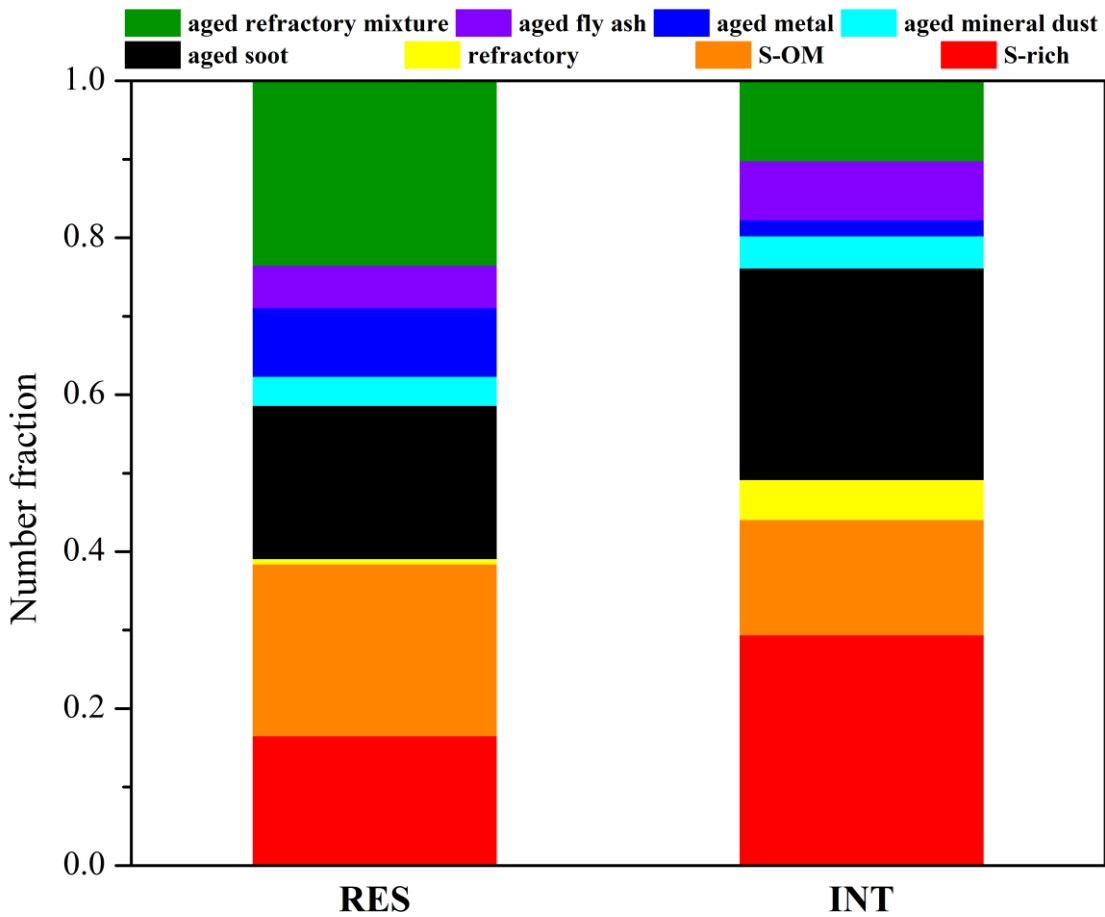


**Figure 2. Number fractions of different particle types in the RES and INT of cloud event #2 and #3 measured**

**by TEM/EDS.**

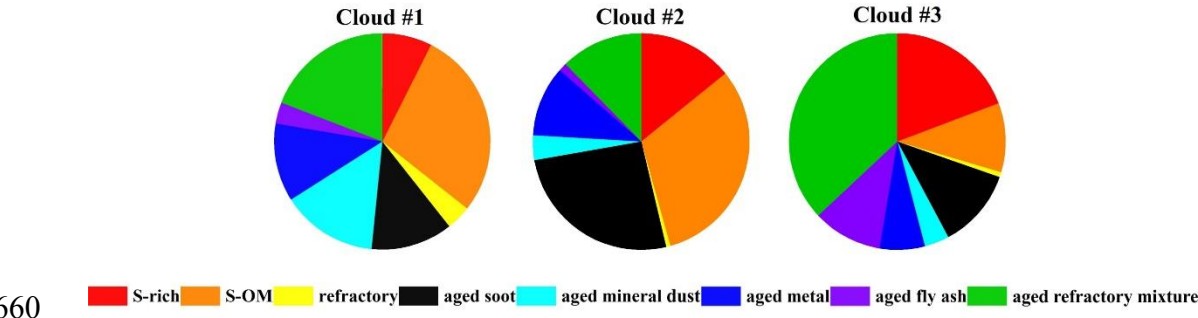

**Figure 3. Number fraction of different particle types in the RES during three cloud events measured by TEM/EDS.**

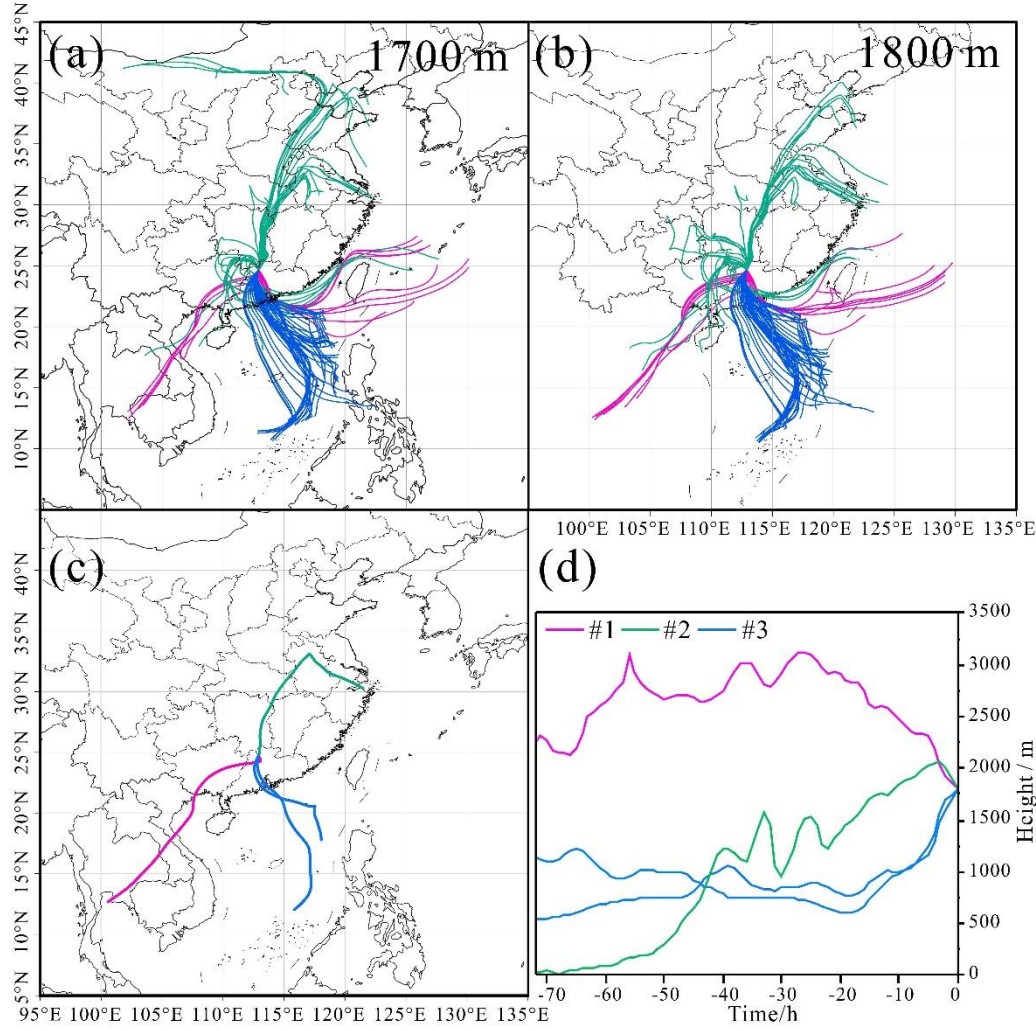

663

**Figure 4. HYSPLIT back trajectories (72 h) for air masses arriving at our sampling site at the height of 1700**

**m (a) and 1800 m (b) hourly during the three cloud events. The HYSPLIT back trajectories at the height of**

**1800 m during sampling periods (c) and heights (above sea level) of the air masses during transport (d). The**

**horizontal axis represents several time points (0-72 h) before the time point input into the HYSPLIT model.**

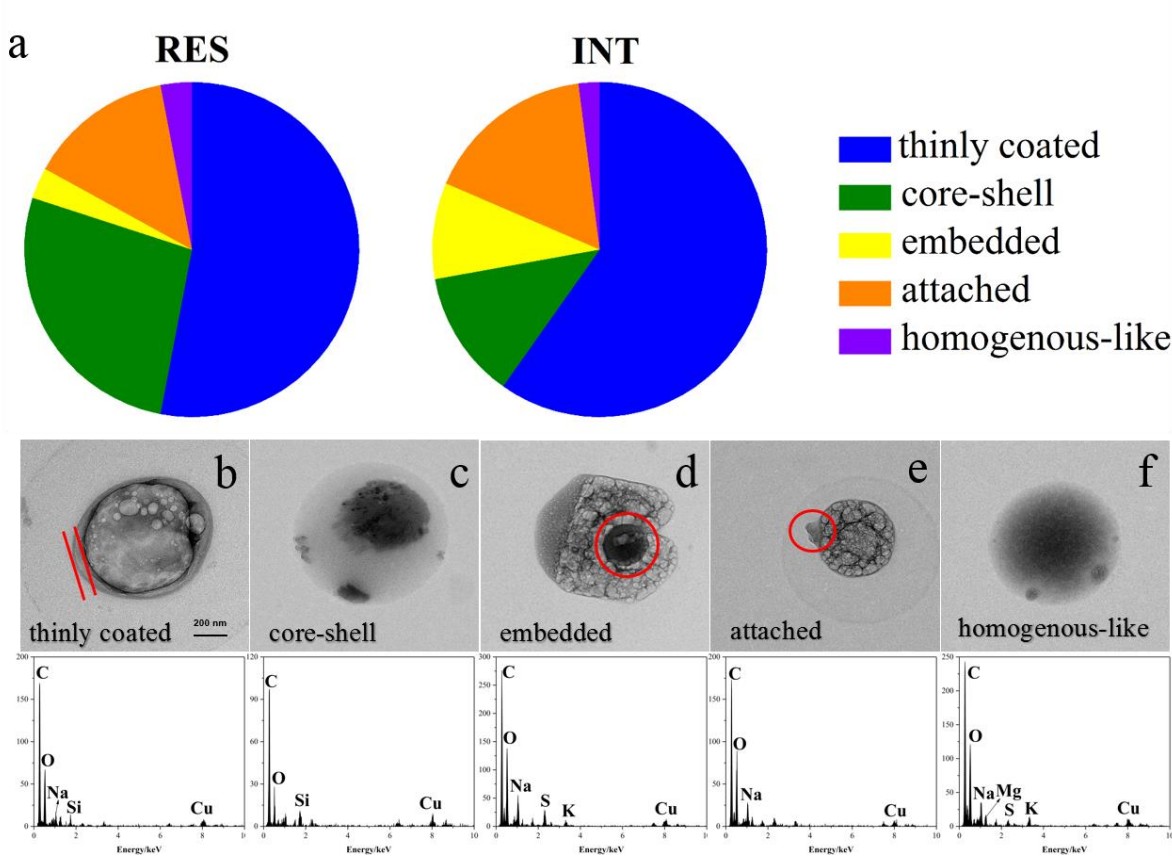


**Figure 5. Number fractions of OM-containing particles with different mixing structures in the RES and INT (a) and**
**typical TEM images and corresponding EDS spectra of OM: thinly coated (b); core-shell (c); embedded (d); attached**
**(e); homogenous-like (f) during cloud event #2 and #3.**

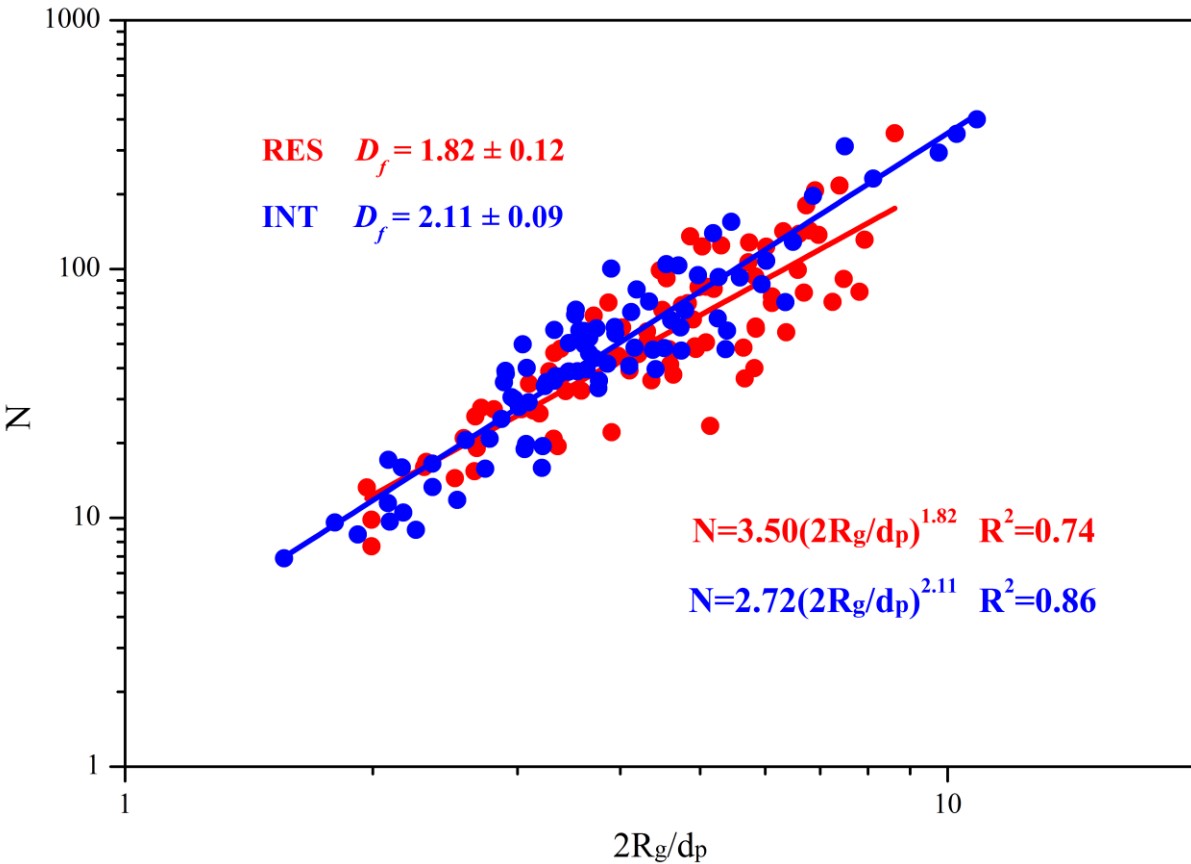


**Figure 6. Fractal dimensions of soot in the RES and INT during cloud event #2 and #3.**