# Peer review of "Impact of in-cloud aqueous processes on the chemical"

_Atmospheric Chemistry and Physics, 2020_

## Referee Comment (RC1) · Anonymous Referee #1 · 26 May 2020

In this paper, the authors did a good job of presenting their results on the different morphologies and mixing states of activated and interstitial particles. I think that the analysis could be strengthened by further exploring the connection between the offline TEM-EDS analysis and the online SPAMS analysis. As Prof. Surratt mentioned, it has been previously shown that SOA can form via in-cloud processes and the formation of SOA can result in core-shell phase morphology. As such, it would be beneficial to assess the SPAMS mass spectra for SOA products, especially given the increase in the number fraction of S-OM relative to S-rich particles in the RES v. INT particles. Additionally, the size distribution of different particle morphologies and mixing states could be assessed and compared to the size-resolved mass spectra from the SPAMS.

[Figure]

Additionally, I would recommend that the authors include the EDS spectra that correspond to the TEM images presented in Figure 3. In the SI, to clarify the difference between figures S1 and S3, I would recommend that the authors change the title of S1 to indicate that the fraction types correspond to EDS.

---

## Referee Comment (RC2) · Anonymous Referee #2 · 10 Jul 2020

General:

This paper compares single-particle measurements of particle size, composition, and morphology of in-cloud and interstitial particles collected at a rural site in China. This paper mainly comments on the role of aqueous chemistry in forming organic shells and the observation of more branched soot particles in cloud. I have several comments about this work to be considered before publication.

Major Comments:

1. A lot of the most important details and figures are in the SI rather than in the main text. Also, a lot of the main supporting data came from SPAMS analysis which was not

described in the methods.

2. The argument regarding core-shell and liquid-liquid phase separations was a bit confusing since previous work has shown that the core-shell morphology can break down as RH is increased.

Specific Comments:

Introduction:

1. Lines 59-60: Nitric oxide is a gas, not particulate matter.

2. Line 63: not sure how "decomposed" is being used in this sentence.

3. Line 76: reword "this process might not be neglected"

4. Line 78: Also see [Moffet and Prather, 2009]

5. Lines 77-86: please also comment on the finding that organic coatings caused the collapse of soot particles from [Spencer and Prather, 2006]

Methods:

1. Lines 100-101: What is meant by "almost unaffected by local anthropogenic sources"?

2. Lines 117-120: I think that Table S1 and the air mass back trajectories should be shown in the main paper. It will help give context for what was different between the different cloud events to help interpret the results.

3. Line 125: change "folds" to "fold".

4. Line 128: change "vacuumed" to "vacuum" and define NH4NO3.

5. I couldn't follow the methodology given in section 2.4. Please add more details.

6. A lot of SPAMS data is brought in to corroborate the results. I suggest that details on the SPAMS needs to be added to the methods if the data is being used.

Results:

1. Line 153: some fresh soot particles can have sulfate, see [Moffet and Prather, 2009].

2. Lines 154-156: the methods for identifying each component should be moved from the SI to the methods section of the paper.

3. Lines 159-161: I would think it would be important to explicitly detail the mixture for your results. I found these classifications really confusing and hard to keep straight.

4. Lines 167-168: I recommend bringing Figure S1 into the main paper.

5. Lines 168-169: Not sure what is meant by "influenced by air masses". More description of the different conditions and air mass conditions encountered for each cloud event will help the authors interpret their single particle findings.

6. Lines 171-172: What is meant by "as confirmed by SPAMS data"?

7. Line 176: what is meant by "part of"? Can this be made more quantitative?

8. Figure 3, should "coating" be "thin coating" instead to better distinguish the morphology?

9. I found the coating thickness definitions to be confusing especially because they overlap. I'm not quite sure how the coating thickness was used to robustly distinguish particles classified as "coating" vs "core-shell"

10. Lines 219-222 imply that the site is polluted, but the site was presented as a background site.

11. Line 223: reword "follow up strong interactions" to "heterogeneous and multiphase reactions"

12. Lines 225-227 seem to imply that there is more data that was not presented. Please rephrase.

13. I'm very confused as to how the O/C ratios were determined. Perhaps I missed

something, but I thought that the detector used only detected elements heavier than C and it is not clear how the background from the carbon film is accounted for.

14. One of the main conclusions of this paper is regarding oxidized coatings formed via aqueous chemistry, yet the main table showing this is in the SI. I suggest bringing Table S2 into the main paper.

15. The O/C values should be stated in the main paper.

16. Lines 242-245: If ion peak ratios from SPAMS are discussed, then SPAMS must be included in the methods section and the interpretation of the ion peak ratios needs much more interpretation to connect to the data presented in this paper.

17. Line 251: also site [Moffet and Prather, 2009]

18. Lines 254-256: I don't follow the logic regarding non-volatile material and branching. I suggest that the authors more clearly present this argument.

19. Lines 258-260: could this just be showing the role of particle size where unaged soot is larger and more CCN active than smaller, aged particles?

20. Lines 268-270: is there a figure showing the off-center positions of the soot?

References:

Moffet, R. C., and K. A. Prather (2009), In-situ measurements of the mixing state and optical properties of soot with implications for radiative forcing estimates, Proceedings of the National Academy of Sciences of the United States of America, 106(29), 11872-11877.

Spencer, M. T., and K. A. Prather (2006), Using ATOFMS to determine OC/EC mass fractions in particles, Aerosol Science and Technology, 40(8), 585-594.

---

## Author Comment (AC2) · 4 Aug 2020

**Comment from Referee #2:**

General:

This paper compares single-particle measurements of particle size, composition, and morphology of in-cloud and interstitial particles collected at a rural site in China. This paper mainly comments on the role of aqueous chemistry in forming organic shells and the observation of more branched soot particles in cloud. I have several comments about this work to be considered before publication.

We appreciate the constructive suggestions and comments from the referee. The referee's comments are in the black text followed by our response in the blue text as follows.

Major Comments:

1. A lot of the most important details and figures are in the SI rather than in the main text. Also, a lot of the main supporting data came from SPAMS analysis which was not described in the methods.

Thanks for your suggestions. We have moved some tables and figures from the SI to the main text, and we have added more information about the SPAMS in the methods. Please see below for specific responses.

2. The argument regarding core-shell and liquid-liquid phase separations was a bit confusing since previous work has shown that the core-shell morphology can break down as RH is increased.

We agree with your comment. What we want to express is that the oxidized organic matter formed in the cloud processes has a great influence on the mixing structure of organic particles after water evaporation within particles, which is conducive to the existence of organic particles in a core-shell structure after cloud events. Since we did not express clearly, we modified the relevant statements in the manuscript. Please refer

to lines 36-37 and 319-322:

Lines 36-37: The sentence, "Our results highlight the in-cloud formation of more oxidized organic shells on the activated particles.", has been changed to "Our results highlight that the formation of more oxidized organic matter in the cloud contributes to the existence of organic shells after cloud processing.".

Lines 319-322: The sentence, "The prevalence of OM shelled particles upon in-cloud processes also supports a recent laboratory observation depicting that rapid film formation and fast heterogeneous oxidation can provide an efficient way of converting water-insoluble organic films into more water-soluble components in aerosols or cloud droplets (Aumann and Tabazadeh, 2008).", has been changed to "The prevalence of OM shelled particles after cloud processing also supports a current laboratory observation depicting that rapid film formation and fast heterogeneous oxidation can provide an efficient way of converting water-insoluble organic films into more water-soluble components in aerosols or cloud droplets (Aumann and Tabazadeh, 2008).".

Specific Comments:

Introduction:

1. Lines 59-60: Nitric oxide is a gas, not particulate matter.

Thank you for pointing out this. In the article we quoted, the authors used "nitric oxide" to represent the $NO^+$ signal measured by ATOFMS, which refers to nitrate. In order to avoid ambiguity, we have changed "nitric oxide" to "nitrate".

2. Line 63: not sure how "decomposed" is being used in this sentence.

We are sorry for the misunderstanding. The word (decomposed) has been removed, and the sentence has been changed to "These results indicate that both RES and INT present complex mixtures, and carbonaceous matter (i.e., organic materials (OM) and soot) is important material in the cloud mass.". Please refer to lines 62-63.

3. Line 76: reword "this process might not be neglected"

The sentence, "this process might not be neglected.", has been changed to "the influence of this process in atmospheric chemistry cannot be neglected.". Please refer to lines 75-76.

4. Line 78: Also see [Moffet and Prather, 2009]

The literature has been cited in the main text. Please refer to lines 77-79:
"For another type of carbonaceous material, soot, there is extensive evidence that the absorption and cloud activation of soot-containing particles can be significantly affected by coatings (Adachi et al., 2010; Wu et al., 2018; Moffet and Prather, 2009)."

5. Lines 77-86: please also comment on the finding that organic coatings caused the collapse of soot particles from [Spencer and Prather, 2006].

The original sentence, "While some studies have found that soot compaction occurs after cloud processing (Bhandari et al., 2019; Ma et al., 2013; Mikhailov et al., 2006), Khalizov et al. (2013) suggested that soot with thin organic coating did not become more compact under high humidity.", has been changed to "While some studies have found that soot restructuring occurs after water processing (Bhandari et al., 2019; Ma et al., 2013; Mikhailov et al., 2006), or being coated by OM (Spencer and Prather, 2006) and sulfate (Zhang et al., 2008), Khalizov et al. (2013) suggested that soot with thin organic coating did not become more compact under high humidity." Please refer to lines 83-86.

Reference:
Spencer, M. T., and Prather, K. A.: Using ATOFMS to determine OC/EC mass fractions in particles, Aerosol Science and Technology, 40, 585-594,

10.1080/02786820600729138, 2006.

Zhang, R., Khalizov, A. F., Pagels, J., Zhang, D., Xue, H., and McMurry, P. H.: Variability in morphology, hygroscopicity, and optical properties of soot aerosols during atmospheric processing, Proceedings of the National Academy of Sciences of the United States of America, 105, 10291-10296, 10.1073/pnas.0804860105, 2008.

Methods:

1. Lines 100-101: What is meant by "almost unaffected by local anthropogenic sources"?

The sampling site is surrounded by a national park forest (273 km$^2$), and there are scarcely any emissions from anthropogenic activities.

2. Lines 117-120: I think that Table S1 and the air mass back trajectories should be shown in the main paper. It will help give context for what was different between the different cloud events to help interpret the results.

We agree with you. Table S1 and Figure S2 (the air mass back trajectories) have been moved to the main text, and are numbered as Table 1 and Figure 4.

3. Line 125: change "folds" to "fold".

It has been changed accordingly.

4. Line 128: change "vacuumed" to "vacuum" and define NH$_4$NO$_3$.

It has been changed accordingly. The sentence has been changed to "In the TEM vacuum chamber, some volatile substances (e.g., ammonium nitrate (NH$_4$NO$_3$) and volatile organic matter) would be lost.". Please refer to lines 129-130.

5. I couldn't follow the methodology given in section 2.4. Please add more details.

We have added detailed calculations in the section 2.4 (that is section 2.5 now). We have now included an introduction on two parameters including $k_a$ and $\alpha$, which depends on the degree of monomer overlap ($\delta$) in the aggregate. Furthermore, we have also supplemented the calculation of $\delta$ herein. Please refer to lines 174-180:

"The value of $k_a$ and $\alpha$ depends on the degree of monomer overlap ($\delta$) in the aggregate (Oh and Sorensen, 1997), and $\delta$ can be determined by:

$$\delta = \frac{2a}{l}$$

where $a$ is monomer radius, and $l$ is the center distance of adjacent monomers. The value of parameters including $a$, $l$, $A_a$, $A_p$, $L_{max}$, and $d_p$ can be obtained by analyzing TEM images. Then $D_f$ can be calculated by the above four formulas."

Reference:

Oh, C., and Sorensen, C. M.: The effect of overlap between monomers on the determination of fractal cluster morphology, Journal of Colloid and Interface Science, 193, 17-25, 10.1006/jcis.1997.5046, 1997.

6. A lot of SPAMS data is brought in to corroborate the results. I suggest that details on the SPAMS needs to be added to the methods if the data is being used.

The detailed information of the SPAMS has been added in the methods (section 2.4). And the identification of particles measured by the SPAMS is provided in the SI.

Results:

1. Line 153: some fresh soot particles can have sulfate, see [Moffet and Prather, 2009].

We agree with your comment. Since there is hardly any anthropogenic source around our sampling site, the collected soot particles are assumed to experience long-distance

transport and complex aging processes. The aging state of the soot particles may also be reflected by the associated more intense sulfate peaks measured by the SPAMS, compared with those observed in urban areas, as discussed in our previous publication (Zhang et al., 2017). Considering that much secondary inorganic matter and organic matter might be generated during the cloud processes, this study distinguishes fresh and aged particles by the presence or absence of secondary inorganic matter (S-rich) and organic matter (OM).

Zhang, G., Lin, Q., Peng, L., Bi, X., Chen, D., Li, M., Li, L., Brechtel, F. J., Chen, J., Yan, W., Wang, X., Peng, P., Sheng, G., and Zhou, Z.: The single-particle mixing state and cloud scavenging of black carbon: a case study at a high-altitude mountain site in southern China, Atmospheric Chemistry and Physics, 17, 14975-14985, 10.5194/acp-17-14975-2017, 2017.

2. Lines 154-156: the methods for identifying each component should be moved from the SI to the methods section of the paper.

The identification of each component has been moved from SI to the methods section of main text (section 2.3), and original sentence of lines 154-156, "The details involving the identification of each component (S, OM, soot, mineral, metal, fly ash) are provided in the Supporting Information.", has been canceled.

3. Lines 159-161: I would think it would be important to explicitly detail the mixture for your results. I found these classifications really confusing and hard to keep straight.

Thanks for your comment. Fresh mixture has been changed to "refractory" in the full text and SI, and the sentence has been changed to "Aged particle types containing two or more refractory components are named as "aged mixture". It is worth noting that "refractory" refers to the refractory particles without S-rich and OM.". Please refer to lines 188-190.

4. Lines 167-168: I recommend bringing Figure S1 into the main paper.

Thanks for your suggestion. Figure S1 has been moved to the main text, and is numbered as Figure 3.

5. Lines 168-169: Not sure what is meant by "influenced by air masses". More description of the different conditions and air mass conditions encountered for each cloud event will help the authors interpret their single particle findings.

To make it clear, "influenced by air masses" has been removed, and we moved the content about the influence of air masses on the distribution of particle types in the RES in the SI to the main text. Please refer to Lines 200-212:

   "The different air masses are expected to affect the distribution of particle types. The distribution of several types of particles in the RES were observed to be divergent in different cloud events, corresponding to different air masses, as shown in Figure 3 and Figure 4. The number fraction of OM-containing particles was the highest (81%) in cloud event #2, which might be partly attributed to the higher concentration of O3 during cloud event #2 (Table S1). And the samples of cloud event #2 sampled at noon. Higher solar radiation at the sampling time might also promote heterogeneous photochemical oxidation reactions during the cloud process and increased the generation of OM within cloud droplets (Xu et al., 2017). Aged metal particles accounted a similar percentage (7-12%) for three cloud events. The proportion of aged mineral during cloud event #1 (14%) was nearly four times those in the other two cloud events. Aged fly ash particles had the highest proportion (10%) in cloud event #3 compared with the other two cloud events, which is most probably influenced by the different air masses (Figure 4). Aged mineral particles of cloud event #1 may be influenced by the long-distance transportation of dust from Southeast Asia (Salam et al., 2003). Clearly, aged fly ash particles of cloud event #3 are associated with the air

masses from the PRD region with a dense distribution of industrial facilities there (Cao et al., 2006)."

6. Lines 171-172: What is meant by "as confirmed by SPAMS data"?

That sentence has been removed in the revised manuscript.

7. Line 176: what is meant by "part of"? Can this be made more quantitative?

Since aged refractory and aged mixture particles include S/S-OM/OM-refractory and S/S-OM/OM-soot/mineral/metal/fly ash. OM-containing particles refer to S-OM/OM-refractory and S-OM/OM-soot/mineral/metal/fly ash, which is part of aged refractory and aged mixture particles. In the RES and INT during cloud event #2 and #3, OM-containing particles account for 63% and 32% of the aged refractory particles and 51% and 43% of the aged mixture particles.

8. Figure 3, should "coating" be "thin coating" instead to better distinguish the morphology?

Thanks for your suggestion. We have changed the mixing structure "coating" to "thinly coated" in the full text and SI.

9. I found the coating thickness definitions to be confusing especially because they overlap. I'm not quite sure how the coating thickness was used to robustly distinguish particles classified as "coating" vs "core-shell".

The electron beam of TEM penetrates the particle, and the internal structure of the particle can be observed (Li et al., 2016). Thus, although the organic matter wraps the internal material, the organic coating and shell can still be easily identified. The major difference between "coating (that is thinly coated now)" and "core-shell" is the relative

thickness of organic coating and shell in the particle is. Core-shell structure possessed thicker organics than thinly coated structure, and the thickness of OM-coating and OM-shell is 12-150 nm and 86-2110 nm in this study, respectively. Please refer to Lines 218-224.

Reference:

Li, W., Sun, J., Xu, L., Shi, Z., Riemer, N., Sun, Y., Fu, P., Zhang, J., Lin, Y., Wang, X., Shao, L., Chen, J., Zhang, X., Wang, Z., and Wang, W.: A conceptual framework for mixing structures in individual aerosol particles, Journal of Geophysical Research-Atmospheres, 121, 13784-13798, 10.1002/2016jd025252, 2016.

10. Lines 219-222 imply that the site is polluted, but the site was presented as a background site.

We are sorry for the misunderstanding. As we described, the results of this study are similar to those of unpolluted remote areas, but different from those in polluted areas. The original sentence has been changed from "Such a mixing structure is similar to those observed in the Arctic, background, or rural atmosphere (Hiranuma et al., 2013; Li et al., 2016; Yu et al., 2019), but is different from previous findings in polluted air where OM is typically mixed with sulfate (Li et al., 2016)." to "Such a mixing structure is similar to those observed in the Arctic, background, or rural atmosphere (Hiranuma et al., 2013; Li et al., 2016; Yu et al., 2019), but is different from other findings in polluted areas where OM is typically mixed with sulfate (Li et al., 2016)."

11. Line 223: reword "follow up strong interactions" to "heterogeneous and multiphase reactions"

It has been changed accordingly.

12. Lines 225-227 seem to imply that there is more data that was not presented. Please

rephrase.

Thanks for pointing out this. The sentence has been changed to "Recently, Gorkowski et al. (2020) came up with a particle morphology prediction framework developed for mixtures of organic aerosol based on the measurements from aerosol optical tweezers experiments and literature data, and they hypothesized the core-shell morphology dominated by secondary organic aerosols (SOA) in the shell phase.". Please refer to lines 262-265.

13. I'm very confused as to how the O/C ratios were determined. Perhaps I missed something, but I thought that the detector used only detected elements heavier than C and it is not clear how the background from the carbon film is accounted for.

The O/C value is measured by EDS (energy-dispersive X-ray spectrometry), which can obtain the weight and atomic number proportion of elements heavier than carbon (Z $\geqslant$ 6), and some limitations of O/C have also been expressed in the main text. Please refer to lines 266-271:
"It should be noted that the O/C ratio of organic coating and shell is underestimated herein due to the copper grid evenly covered by carbon film. And, while some loss of volatile organic compounds during the TEM/EDS analysis may affect the O/C of particles, the relatively higher O/C ratio for the RES is still affirmative. Droplets are expected to dissolve more abundance of volatile organic compounds (Chakraborty et al., 2016), evaporation of which would result in an underestimate of O/C to a higher degree rather than the INT."

14. One of the main conclusions of this paper is regarding oxidized coatings formed via aqueous chemistry, yet the main table showing this is in the SI. I suggest bringing Table S2 into the main paper.

Thanks for your suggestion. Table S2 has been moved to the main text, and numbered as Table 2.

15. The O/C values should be stated in the main paper.

The O/C value of organic shell and coating has been in the main text. Please refer to lines 271-274:
"We found that the average value of the O/C ratio of RES is higher than INT, and the average value of the O/C ratio of RES with core-shell structure is 0.23, which is two times that with thinly coated structure (0.11) (Table 2), indicating that these RES with core-shell particles are more oxidized."

16. Lines 242-245: If ion peak ratios from SPAMS are discussed, then SPAMS must be included in the methods section and the interpretation of the ion peak ratios needs much more interpretation to connect to the data presented in this paper.

We agree with your comment. The operating principle of the SPAMS has been added in the method section (section 2.4), and the introduction of the relative peak area is also described, that is "The relative peak area of characteristic peaks of specific material in the mass spectra is generally applied to indicate its relative abundance in the particle.".

17. Line 251: also site [Moffet and Prather, 2009]

It has been added in the text. Please refer to line 289.
Lines 288-290: While some previous studies demonstrated that soot aggregates tend to be more compact (with larger $D_f$) after aging or cloud processing (Adachi and Buseck, 2013; Wu et al., 2018; Moffet and Prather, 2009), our results suggest that in-cloud processes may result in more branched soot, as shown in Figure 6.

Reference:

Moffet, R. C., and Prather, K. A.: In-situ measurements of the mixing state and optical properties of soot with implications for radiative forcing estimates, Proceedings of the National Academy of Sciences of the United States of America, 106, 11872-11877, 10.1073/pnas.0900040106, 2009.

18. Lines 254-256: I don't follow the logic regarding non-volatile material and branching. I suggest that the authors more clearly present this argument.

Thank you for pointing out this. We have described that in detail. The sentences have been changed to "The first and the most likely reason is that some of the soot aggregates are immediately encapsulated by non-volatile materials (such as organic matter) after emission by combustion sources. These coatings fill the spaces between the branches of soot aggregates, which inhibits the relatively large deformation and reconfiguration of the soot aggregates during transport and activation into cloud droplets (Zhang et al., 2018). Differently, soot aggregates may shrink easily and become more compact during long-distance transport, if the soot aggregates are emitted without non-volatile coatings (Adachi and Buseck, 2013)." Please refer to lines 292-298.

19. Lines 258-260: could this just be showing the role of particle size where unaged soot is larger and more CCN active than smaller, aged particles?

We are sorry for the misunderstanding. We described the $D_f$ (fractal dimension) of soot particles in the RES and INT in the section 3.2, which is $1.82 \pm 0.05$ and $2.16 \pm 0.05$, respectively. And here, we described the ECD (equivalent circle diameter) of soot particles in the RES and INT, which is 266 nm and 247 nm. So, compared with soot in the RES, soot in the INT particles have larger $D_f$ and smaller ECD. To be more clear, the sentence has been changed to "We show that soot aggregates have higher $D_f$ and lower average ECD in the INT (247 nm) than in the RES (266 nm), which means that larger, less dense soot particles are easier to act as CCN.". Please refer to lines 298-300.

20. Lines 268-270: is there a figure showing the off-center positions of the soot?

Yes. The Figure S4 (a, b, c, g) show the off-center positions of the soot. "Figure S4" is added in the sentence. Please refer to lines 307-309:

"Our observations at the background site show that the majority of soot aggregates in both RES and INT (~80%) are located in off-center positions (Figure S4), having less compact shapes even after being coated."

References:

Moffet, R. C., and K. A. Prather (2009), In-situ measurements of the mixing state and optical properties of soot with implications for radiative forcing estimates, Proceedings of the National Academy of Sciences of the United States of America, 106(29), 11872-11877.

Spencer, M. T., and K. A. Prather (2006), Using ATOFMS to determine OC/EC mass fractions in particles, Aerosol Science and Technology, 40(8), 585-594.

---

## Author Response (AR1)

**Comment from the editor:**

Dear Authors,

Thank you for addressing my comments at this stage. I will go ahead and start the formal review process (ACPD). I did want to point out that your Figure 1 in the reply has a mass spectrum similar to Figure 1 from Hatch et al. (2011, ES&T). Hatch et al. saw a small fraction as well, but is this due to the ionization efficiency being low? Meaning, even though the ion intensity is low, do you really know if that means it is low in abundance in the RES? Please consider that comment during the review process.

Thanks for your handling and comments to further improve our manuscript. It is possible that the ionization efficiency may lead to the small fraction of organosulfate, however, there is still no data to support this. Thus, we cannot confirm that it is low in abundance in the RES, since the SPAMS only provides the relative intensity of ion peaks, which can be used to indicate the relative abundance of a species in a particle. However, we observed both the small number fraction of particles containing organosulfate and extremely low relative intensity for organosulfate may still reflect the limited occurrence of organosulfate in the RES.

**Comment from Referee #1:**
In this paper, the authors did a good job of presenting their results on the different morphologies and mixing states of activated and interstitial particles. I think that the analysis could be strengthened by further exploring the connection between the offline TEM-EDS analysis and the online SPAMS analysis.

We would like to thank the referee for the positive and valuable comments to improve our manuscript. We agree with the comments and have strengthened the analysis, in particular, the connection between the TEM/EDS and the SPAMS results. As suggested, the size distribution and mass spectral information were included to support the discussion. We have addressed the specific comments in the sections below and made the appropriate revisions to the manuscript. The referee's comments are in the black text followed by our response in the blue text.

As Prof. Surratt mentioned, it has been previously shown that SOA can form via in-cloud processes and the formation of SOA can result in core-shell phase morphology. As such, it would be beneficial to assess the SPAMS mass spectra for SOA products, especially given the increase in the number fraction of S-OM relative to S-rich particles in the RES v. INT particles.

Thanks for your comment. Indeed, we also hope to obtain more information about OM from SPAMS to explain and corroborate the results obtained from TEM/EDS. In the section 3.3, we use the ratios of relative peak area of organics to sulfate of OM particles during in-cloud (RES and INT) and pre-cloud (Ambient) periods from the data of SPAMS (Table S2), to help explain the in-cloud formation of OM found by TEM/EDS, which show that the ratios of relative peak area between organics and sulfate are similar between the INT and particles before cloud event, whereas they are higher in the RES.

Additionally, the size distribution of different particle morphologies and mixing states could be assessed and compared to the size-resolved mass spectra from the SPAMS.

Thanks for your comment. We have added the size-resolved number fraction distributions of the RES and INT by TEM/EDS (Figure S8) and SPAMS (Figure S9). We also added simple description and comparison in the SI:

The size distribution of different particle types revealed that S-rich and aged soot particles were predominant in smaller size segments, and aged mixture particles in larger size segments (Figure S8). Likewise, the size-resolved number fractions of different particle types from the results of the SPAMS also showed that the BC-containing particles were mainly distributed between 0.1 and 1.3 μm, representing ~80% of the submicron RES and ~73% of the submicron INT population, respectively (Figure S9).

[Figure]

Figure S8. Size-resolved number fraction distributions of RES and INT by TEM/EDS.

[Figure]

Figure S9. Size-resolved number fraction distributions of RES and INT by SPAMS.

Additionally, I would recommend that the authors include the EDS spectra that correspond to the TEM images presented in Figure 3.

Thanks for your suggestion. We have added the EDS spectra of the OM-containing particles corresponding to each TEM image in Figure 3 (that is Figure 5 now):

[Figure]

Figure 5. Number fractions of the OM-containing particles with different mixing structures in the RES and INT (a) and typical TEM images and corresponding EDS spectra of OM: thinly coated (b); core-shell (c); embedded (d); attached (e); homogenous-like (f) during cloud event #2 and #3.

In the SI, to clarify the difference between figures S1 and S3, I would recommend that the authors change the title of S1 to indicate that the fraction types correspond to EDS.

Thanks for pointing out this. We have changed the title of Figure S1 (that is Figure 3 now) from "Number fraction of different particle types in the RES during three cloud events." to "Number fraction of different particle types in the RES during three cloud events measured by TEM/EDS.".

**Comment from Referee #2:**

General:

This paper compares single-particle measurements of particle size, composition, and morphology of in-cloud and interstitial particles collected at a rural site in China. This paper mainly comments on the role of aqueous chemistry in forming organic shells and the observation of more branched soot particles in cloud. I have several comments about this work to be considered before publication.

We appreciate the constructive suggestions and comments from the referee. The referee's comments are in the black text followed by our response in the blue text as follows.

Major Comments:

1. A lot of the most important details and figures are in the SI rather than in the main text. Also, a lot of the main supporting data came from SPAMS analysis which was not described in the methods.

Thanks for your suggestions. We have moved some tables and figures from the SI to the main text, and we have added more information about the SPAMS in the methods. Please see below for specific responses.

2. The argument regarding core-shell and liquid-liquid phase separations was a bit confusing since previous work has shown that the core-shell morphology can break down as RH is increased.

We agree with your comment. What we want to express is that the oxidized organic matter formed in the cloud processes has a great influence on the mixing structure of organic particles after water evaporation within particles, which is conducive to the existence of organic particles in a core-shell structure after cloud events. Since we did not express clearly, we modified the relevant statements in the manuscript. Please refer to lines 36-37 and 319-322:

Lines 36-37: The sentence, "Our results highlight the in-cloud formation of more oxidized organic shells on the activated particles.", has been changed to "Our results highlight that the formation of more oxidized organic matter in the cloud contributes to the existence of organic shells after cloud processing.".

Lines 319-322: The sentence, "The prevalence of OM shelled particles upon in-cloud processes also supports a recent laboratory observation depicting that rapid film formation and fast heterogeneous oxidation can provide an efficient way of converting water-insoluble organic films into more water-soluble components in aerosols or cloud droplets (Aumann and Tabazadeh, 2008).", has been changed to "The prevalence of OM shelled particles after cloud processing also supports a current laboratory observation depicting that rapid film formation and fast heterogeneous oxidation can provide an efficient way of converting water-insoluble organic films into more water-soluble components in aerosols or cloud droplets (Aumann and Tabazadeh, 2008).".

Specific Comments:

Introduction:

1. Lines 59-60: Nitric oxide is a gas, not particulate matter.

Thank you for pointing out this. In the article we quoted, the authors used "nitric oxide" to represent the $NO^+$ signal measured by ATOFMS, which refers to nitrate. In order to avoid ambiguity, we have changed "nitric oxide" to "nitrate".

2. Line 63: not sure how "decomposed" is being used in this sentence.

We are sorry for the misunderstanding. The word (decomposed) has been removed, and the sentence has been changed to "These results indicate that both RES and INT present complex mixtures, and carbonaceous matter (i.e., organic materials (OM) and soot) is important material in the cloud mass.". Please refer to lines 62-63.

3. Line 76: reword "this process might not be neglected"

The sentence, "this process might not be neglected.", has been changed to "the influence of this process in atmospheric chemistry cannot be neglected.". Please refer to lines 75-76.

4. Line 78: Also see [Moffet and Prather, 2009]

The literature has been cited in the main text. Please refer to lines 77-79:
"For another type of carbonaceous material, soot, there is extensive evidence that the absorption and cloud activation of soot-containing particles can be significantly affected by coatings (Adachi et al., 2010; Wu et al., 2018; Moffet and Prather, 2009)."

5. Lines 77-86: please also comment on the finding that organic coatings caused the collapse of soot particles from [Spencer and Prather, 2006].

The original sentence, "While some studies have found that soot compaction occurs after cloud processing (Bhandari et al., 2019; Ma et al., 2013; Mikhailov et al., 2006), Khalizov et al. (2013) suggested that soot with thin organic coating did not become more compact under high humidity.", has been changed to "While some studies have found that soot restructuring occurs after water processing (Bhandari et al., 2019; Ma et al., 2013; Mikhailov et al., 2006), or being coated by OM (Spencer and Prather, 2006) and sulfate (Zhang et al., 2008), Khalizov et al. (2013) suggested that soot with thin organic coating did not become more compact under high humidity." Please refer to lines 83-86.

Reference:
Spencer, M. T., and Prather, K. A.: Using ATOFMS to determine OC/EC mass fractions in particles, Aerosol Science and Technology, 40, 585-594,

10.1080/02786820600729138, 2006.

Zhang, R., Khalizov, A. F., Pagels, J., Zhang, D., Xue, H., and McMurry, P. H.: Variability in morphology, hygroscopicity, and optical properties of soot aerosols during atmospheric processing, Proceedings of the National Academy of Sciences of the United States of America, 105, 10291-10296, 10.1073/pnas.0804860105, 2008.

Methods:

1. Lines 100-101: What is meant by "almost unaffected by local anthropogenic sources"?

The sampling site is surrounded by a national park forest (273 km$^2$), and there are scarcely any emissions from anthropogenic activities.

2. Lines 117-120: I think that Table S1 and the air mass back trajectories should be shown in the main paper. It will help give context for what was different between the different cloud events to help interpret the results.

We agree with you. Table S1 and Figure S2 (the air mass back trajectories) have been moved to the main text, and are numbered as Table 1 and Figure 4.

3. Line 125: change "folds" to "fold".

It has been changed accordingly.

4. Line 128: change "vacuumed" to "vacuum" and define NH$_4$NO$_3$.

It has been changed accordingly. The sentence has been changed to "In the TEM vacuum chamber, some volatile substances (e.g., ammonium nitrate (NH$_4$NO$_3$) and volatile organic matter) would be lost.". Please refer to lines 129-130.

5. I couldn't follow the methodology given in section 2.4. Please add more details.

We have added detailed calculations in the section 2.4 (that is section 2.5 now). We have now included an introduction on two parameters including $k_a$ and $\alpha$, which depends on the degree of monomer overlap ($\delta$) in the aggregate. Furthermore, we have also supplemented the calculation of $\delta$ herein. Please refer to lines 174-180:

"The value of $k_a$ and $\alpha$ depends on the degree of monomer overlap ($\delta$) in the aggregate (Oh and Sorensen, 1997), and $\delta$ can be determined by:

$$\delta = \frac{2a}{l}$$

where $a$ is monomer radius, and $l$ is the center distance of adjacent monomers. The value of parameters including $a$, $l$, $A_a$, $A_p$, $L_{max}$, and $d_p$ can be obtained by analyzing TEM images. Then $D_f$ can be calculated by the above four formulas."

Reference:

Oh, C., and Sorensen, C. M.: The effect of overlap between monomers on the determination of fractal cluster morphology, Journal of Colloid and Interface Science, 193, 17-25, 10.1006/jcis.1997.5046, 1997.

6. A lot of SPAMS data is brought in to corroborate the results. I suggest that details on the SPAMS needs to be added to the methods if the data is being used.

The detailed information of the SPAMS has been added in the methods (section 2.4). And the identification of particles measured by the SPAMS is provided in the SI.

Results:

1. Line 153: some fresh soot particles can have sulfate, see [Moffet and Prather, 2009].

We agree with your comment. Since there is hardly any anthropogenic source around our sampling site, the collected soot particles are assumed to experience long-distance transport and complex aging processes. The aging state of the soot particles may also be reflected by the associated more intense sulfate peaks measured by the SPAMS, compared with those observed in urban areas, as discussed in our previous publication (Zhang et al., 2017). Considering that much secondary inorganic matter and organic matter might be generated during the cloud processes, this study distinguishes fresh and aged particles by the presence or absence of secondary inorganic matter (S-rich) and organic matter (OM).

Zhang, G., Lin, Q., Peng, L., Bi, X., Chen, D., Li, M., Li, L., Brechtel, F. J., Chen, J., Yan, W., Wang, X., Peng, P., Sheng, G., and Zhou, Z.: The single-particle mixing state and cloud scavenging of black carbon: a case study at a high-altitude mountain site in southern China, Atmospheric Chemistry and Physics, 17, 14975-14985, 10.5194/acp-17-14975-2017, 2017.

2. Lines 154-156: the methods for identifying each component should be moved from the SI to the methods section of the paper.

The identification of each component has been moved from SI to the methods section of main text (section 2.3), and original sentence of lines 154-156, "The details involving the identification of each component (S, OM, soot, mineral, metal, fly ash) are provided in the Supporting Information.", has been canceled.

3. Lines 159-161: I would think it would be important to explicitly detail the mixture for your results. I found these classifications really confusing and hard to keep straight.

Thanks for your comment. Fresh mixture has been changed to "refractory" in the full text and SI, and the sentence has been changed to "Aged particle types containing two or more refractory components are named as "aged mixture". It is worth noting that "refractory" refers to the refractory particles without S-rich and OM.". Please refer to lines 188-190.

4. Lines 167-168: I recommend bringing Figure S1 into the main paper.

Thanks for your suggestion. Figure S1 has been moved to the main text, and is numbered as Figure 3.

5. Lines 168-169: Not sure what is meant by "influenced by air masses". More description of the different conditions and air mass conditions encountered for each cloud event will help the authors interpret their single particle findings.

To make it clear, "influenced by air masses" has been removed, and we moved the content about the influence of air masses on the distribution of particle types in the RES in the SI to the main text. Please refer to Lines 200-212:

  "The different air masses are expected to affect the distribution of particle types. The distribution of several types of particles in the RES were observed to be divergent in different cloud events, corresponding to different air masses, as shown in Figure 3 and Figure 4. The number fraction of OM-containing particles was the highest (81%) in cloud event #2, which might be partly attributed to the higher concentration of O3 during cloud event #2 (Table S1). And the samples of cloud event #2 sampled at noon. Higher solar radiation at the sampling time might also promote heterogeneous photochemical oxidation reactions during the cloud process and increased the generation of OM within cloud droplets (Xu et al., 2017). Aged metal particles accounted a similar percentage (7-12%) for three cloud events. The proportion of aged mineral during cloud event #1 (14%) was nearly four times those in the other two cloud events. Aged fly ash particles had the highest proportion (10%) in cloud event #3 compared with the other two cloud events, which is most probably influenced by the different air masses (Figure 4). Aged mineral particles of cloud event #1 may be influenced by the long-distance transportation of dust from Southeast Asia (Salam et al., 2003). Clearly, aged fly ash particles of cloud event #3 are associated with the air masses from the PRD region with a dense distribution of industrial facilities there (Cao et al., 2006)."

6. Lines 171-172: What is meant by "as confirmed by SPAMS data"?

That sentence has been removed in the revised manuscript.

7. Line 176: what is meant by "part of"? Can this be made more quantitative?

Since aged refractory and aged mixture particles include S/S-OM/OM-refractory and S/S-OM/OM-soot/mineral/metal/fly ash. OM-containing particles refer to S-OM/OM-refractory and S-OM/OM-soot/mineral/metal/fly ash, which is part of aged refractory and aged mixture particles. In the RES and INT during cloud event #2 and #3, OM-containing particles account for 63% and 32% of the aged refractory particles and 51% and 43% of the aged mixture particles.

8. Figure 3, should "coating" be "thin coating" instead to better distinguish the morphology?

Thanks for your suggestion. We have changed the mixing structure "coating" to "thinly coated" in the full text and SI.

9. I found the coating thickness definitions to be confusing especially because they overlap. I'm not quite sure how the coating thickness was used to robustly distinguish particles classified as "coating" vs "core-shell".

The electron beam of TEM penetrates the particle, and the internal structure of the particle can be observed (Li et al., 2016). Thus, although the organic matter wraps the internal material, the organic coating and shell can still be easily identified. The major difference between "coating (that is thinly coated now)" and "core-shell" is the relative thickness of organic coating and shell in the particle is. Core-shell structure possessed thicker organics than thinly coated structure, and the thickness of OM-coating and OM-shell is 12-150 nm and 86-2110 nm in this study, respectively. Please refer to Lines 218-224.

The size distribution of different particle types revealed that S-rich and aged soot particles were predominant in smaller size segments, and aged mixture particles in larger size segments (Figure S8). Likewise, the size-resolved number fractions of different particle types from the results of the SPAMS

also showed that the BC-containing particles were mainly distributed between 0.1 and 1.3 µm, representing ~80% of the submicron RES and ~73% of the submicron INT population, respectively (Figure S9).

**3 Identification of several types of particles within RES and INT measured by the SPAMS**

All particles with bipolar mass spectra and the size range of $d_{va}$ 0.1–1.9 µm were classified several clusters by an adaptive resonance theory neural network (ART-2a) with a learning rate of 0.05, a vigilance factor of 0.8 and 20 iterations, and merged similar clusters manually. Ten characteristic particle types (Figure S1) were obtained including BC (black carbon)-containing, OC (organic carbon), HMOC

(highly molecular organic carbon), Dust, K-rich, Metal, Na-K, Amines, SS (sea salt) and Others. BC- containing particles are characterized by elemental carbon cluster ions ($m/z$ $12C^{\pm}$, $24C_2^{\pm}$, $36C_3^{\pm}$,

$48C_4^{\pm}$, …). OC particles mainly contain fragment ions of organics ($m/z$ $27C_2H_3^+$, $37C_3H^+$, $43C_2H_3O^+$, -

$26CN^-$, …). The mass spectra of HMOC particles show the presence of peaks of OC particles and some other organic peaks (such as $m/z$ $77C_6H_5^+$, $91C_7H_7^+$). Furthermore, HMOC particles are distinguished from OC particles by marked ion fragments detected in range of $m/z$ > 100. Dust particles present significant ions at $m/z$ $27Al^+$, $40Ca^+$ and $56CaO^+/Fe^+$. K-rich particles are identified according to the strong signal at $m/z$ $39K^+$ only in positive mass spectra. Metal particles show the presence of metal ion peaks (such as $Fe^+$ ($m/z$ 54 and 56), $Mn^+$ ($m/z$ 55), $Pb^+$ ($m/z$ 206, 207 and 208)) in positive mass spectra.

Na-K particles are characterized by peaks at $m/z$ $23Na^+$, $39K^+$, and less intense peaks at $m/z$ $-46NO_2^-$, -

$62NO_3^-$, $-97HSO_4^-$. The mass spectra of amines particles contain ions signals at $m/z$ $59N(CH_3)_3^+$,

$86C_5H_{12}N^+$, $101C_6H_{15}N^+$. SS particles are mainly composed of ions peaks at $m/z$ $23Na^+$, $46Na_2^+$, $62Na_2O^+$,

$63Na_2OH^+$ and $81Na_2Cl^+$. Most particles are observed to internally mixed with sulfate and nitrate ($m/z$ -

46, -62, -97). Particles with unconspicuous mass spectrum characteristics are named others. Specific classification criteria were described in detail elsewhere (Zhang et al., 2015).

[Figure]

**Figure S1. Average positive and negative mass spectra of main nine types particles (BC-containing, OC,**

**HMOC, Dust, K-rich, Metal, Na-K, Amines, SS) measured by SPAMS.**

(a) Cloud #2-RES

(b) Cloud #3-RES

[Figure]

**Figure S2. The chemical composition of RES during cloud event #2 and #3 measured by SPAMS.**

[Figure]

**Figure S3. Time series of chemical composition of RES and INT during sampling periods of cloud event #2**

**and #3 measured by SPAMS.**

[Figure]

**Figure S4. Typical TEM images of soot particles in the RES (a-d) and INT (e-h).**

[Figure]

**Figure S5. The size distribution of S-rich and S-OM particles. There are few S-rich particles with the size of**

**less than 0.2 μm, and the median size are 0.56 μm and 0.76 μm for S-rich and S-OM particles, respectively.**

[Figure]

**Figure S6. Size distribution of RES and INT during cloud event #2 and #3. There are more INT particles**

**when the size is less than 0.8 μm, and more RES particles when the size is larger than 0.8 μm.**

[Figure]

**Figure S7. Average positive and negative mass spectra of OM particles (OC and HMOC) of RES and INT**

**particles during cloud event #2 and #3 measured by SPAMS.**

[Figure]

**Figure S8. Size-resolved number fraction distributions of RES and INT by TEM/EDS.**

[Figure]

**Figure S9. Size-resolved number fraction distributions of RES and INT by SPAMS.**

**Table S1.** The concentration of $NO_X$, $SO_2$, $O_3$, $PM_{10}$ and $PM_{2.5}$ during three cloud events.

| cloud event | $NO_X$ (ppb) | $SO_2$ (ppb) | $O_3$ (ppb) | $PM_{10}$ ($\mu g\ m^{-3}$) | $PM_{2.5}$ ($\mu g\ m^{-3}$) |
|---|---|---|---|---|---|
| #1 | 2.6 | 0.4 | 30.5 | 3.6 | 1.1 |
| #2 | 3.5 | 1.2 | 39.1 | 4.8 | 1.9 |
| #3 | 4.3 | 0.6 | 34.4 | 11.4 | 4.7 |

**Table S2.** The ratios of relative peak area between organics ($m/z$ 27, 29, 37, 43, 50, 51, 61, 63) and sulfate ($m/z$ -97) of OM particles (OC and HMOC) during in-cloud (RES and INT) and pre-cloud (Ambient)

periods.

| | RES | INT | Ambient |
|---|---|---|---|
| Organics/Sulfates | 1.676 | 1.566 | 1.594 |

**Table S3.** Morphological descriptors of soot particles within RES and INT.

| parameters | $A_p$ | $d_p$ | $L_{max}$ | $N$ | $D_f$ | $k_g$ |
|---|---|---|---|---|---|---|
| RES | 1658(175) | 43(2) | 255(12) | 66(8) | 1.82(0.12) | 3.5(0.08) |
| INT | 1842(133) | 46(2) | 316(16) | 68(6) | 2.11(0.09) | 2.72(0.05) |

$A_p$, mean projected area of the monomer; $d_p$, monomer diameter; $L_{max}$, maximum length of soot aggregates; $N$, number of monomers in a soot aggregate; $D_f$, mass fractal dimension; $k_g$, structural coefficient. In parentheses are the standard error of $A_p$, $d_p$, $L_{max}$, $N$, $D_f$ and $k_g$.

**Table S4.** Overlap ($\delta$), constant ($k_a$) and empirical exponent ($\alpha$).

| parameters | $\delta$ | $k_a$ | $\alpha$ |
|:---:|:---:|:---:|:---:|
| RES | 1.54 | 1.52 | 1.13 |
| INT | 1.4 | 1.44 | 1.11 |

---

## Author Response (AR2)

**Comments from the editor:**

The 2 reviewers that reviewed your original manuscript have had a chance to review the revised manuscript and your replies. I have also examined this discussion, and based on the latest reviewer comments I would like for you to address these latest comments before full publication is considered. Specifically, between the 2 reviewers there are 3 major comments that need to be carefully addressed:

Thanks for your careful examination of our manuscript and allowing us to further revise the manuscript. We highly appreciate your comments and the reviewers' suggestions, which undoubtedly improve our manuscript. Based on these comments and suggestions, we have made careful modifications to the original manuscript.

Major Comments:

1. (Reviewer 1) There are a lot of grammatical errors in this paper. I only list a few here but encourage the authors to carefully review and correct their work prior to resubmission.

We want to thank the reviewer for their careful reviewing of the manuscript. We have carefully checked the full manuscript and revised the possible grammatical errors and/or typos, including those mentioned by reviewer 1#.

2. (Reviewer 1) I suggest that the authors provide more context for what is in the SI. For instance, a lot of the SPAMS description is in the SI. The authors need to point this out in the main text as well.

We agree with the comment. We have provided more context for the SPAMS measurements in the main text as suggested. Please refer to the detailed response below or the section 2.4 of the revised manuscript.

3. (Reviewer 2) I do think that there is still some confusion as the role of ionization efficiency and the complexities in equating concentration to relative intensity in a mass spectrum. I do think this should be clarified before the paper is published.

Thanks for the comment. We have clarified the limitation of using the relative peak area as a proxy for the relative abundance of the measured chemical composition. Please refer to the detailed response below for more details.

In addition to these major comments, please be sure to address all of Reviewer 1's specific comments.

Thank you for reminding. We have adequately addressed all the reviewers' specific comments in the following text. Below we give a detailed response to each of the concerns raised by the reviewers. The reviewers' comments are in the black text, followed by our response in the blue text.

Due to the nature of the comments, once you resubmit I'll carefully review and make a decision. Most likely I won't need to solicit any further comments from the reviewers.

Thanks so much and look forward to your revised manuscript.

Most sincerely, Jason Surratt

**Comments from Review #1**

General:

This paper compares single-particle measurements of particle size, composition, and morphology of in-cloud and interstitial particles collected at a rural site in China. This paper mainly comments on the role of aqueous chemistry in forming organic shells and the observation of more branched soot particles in cloud. The revision is much clearer and improved. I recommend publication after consideration of my comments.

Major Comments:

1. There are a lot of grammatical errors in this paper. I only list a few here but encourage the authors to carefully review and correct their work prior to submission.

Thanks for the comment. We have carefully checked and revised the full manuscript and supporting materials, including errors mentioned in the specific comments.

2. I suggest that the authors provide more context for what is in the SI. For instance, a lot of the SPAMS description is in the SI. The authors need to point this out in the main text as well.

Thanks for the suggestions. We have added a detailed description of the performance of the SPAMS in the section 2.4. Please refer to the detailed response below or the section 2.4 of the revised manuscript.

Specific Comments:

Abstract:

Line 33 and throughout the paper: "mineral" should be changed to "mineral dust"

It has been changed accordingly.

Lines 33-34: I still have some reservations about the "aged mixture" label for the most important particle type for RES. It is a non-descript particle class. I suggest that the authors re-think this classification to appeal to a broader audience.

Thanks for the helpful suggestion. In this study, "aged mixture" refers to the particles containing S-rich or OM and two or more refractory components. We have changed the word "aged mixture" to "aged refractory mixture", which may be appropriate to represent aged particles containing two or more refractory components.

Line 38: I suggest that authors give a very quick definition of the $D_f$ to appeal to a broader audience.

Thanks for the suggestion, the definition of the $D_f$ has been added. Fractal dimension ($D_f$), a morphologic parameter to represent the branching degree of particles, for soot particles in the RES (1.82 ± 0.12) is lower than that in the INT (2.11 ± 0.09), which indicates that in-cloud processes may result in less compact soot.

Introduction:

1. Line 46: remove "at a certain supersaturation"

It has been removed accordingly.

2. Line 49: change "coagulation" to "coalescence"

It has been changed accordingly.

3. I think what is missing is some context for cloud droplet formation and evaporation cycles. I suggest describing the number of cloud/evaporation cycles aerosols go through in their lifetime and that these processes critically shape particle size, morphology, and composition with impacts on clouds and direct radiative forcing.

Thanks for the suggestion. The number of cloud/evaporation cycles experienced by aerosols in their lifetime is quite uncertain, which is affected by atmospheric convection, pressure, water vapor content, aerosol particle size and other factors, so it is difficult to evaluate. In addition, the influence of the change of chemical and physical properties caused by cloud processes on the aerosol and radiative forcing after evaporation has also been summarized, please refer to lines 48-54:

On the other hand, in-cloud processes, including the formation of sulfate, nitrate, and water-soluble organics, and the physical processes such as collision and coalescence, would substantially change the physical and chemical properties of the activated particles (Kim et al., 2019; Ma et al., 2013; Roth et al., 2016; Wu et al., 2013). Given that the morphology and mixing state are vital in determining the optical properties of particles (Adachi et al., 2010; Wu et al., 2018), changes of these properties upon in-cloud processes would further affect the subsequent atmospheric processes (e.g., cloud activation, heterogeneous reactions) and radiative forcing of particles after droplet evaporation.

4. Line 68: incomplete sentence

Thanks for pointing out this. The sentence has been changed to "In particular, physical properties play a leading role in cloud activation of inorganic/organic mixed particles (Topping et al., 2007).". Please refer to lines 67-68.

5. Lines 81-87: make it clear what $D_f$ impacts: size? Radiative impacts?

Thanks for the comment. The impact of $D_f$ on the size and radiation forcing has been described here. Please refer to lines 81-84:

Fractal dimension ($D_f$) is widely used to indicate the extent of branching of soot (Brasil et al., 1999), with densely packed or compacted soot particles having higher $D_f$ than chain-like branched clusters or open structures. When the branched soot particles become compact, their size will decrease, but the scattering cross-section will be greater (Radney et al., 2014; Zhang and Mao, 2020).

Radney, J. G., You, R., Ma, X., Conny, J. M., Zachariah, M. R., Hodges, J. T., and Zangmeister, C. D.: Dependence of Soot Optical Properties on Particle Morphology: Measurements and Model Comparisons, Environmental Science & Technology, 48, 3169-3176, 10.1021/es4041804, 2014.

Zhang, X., and Mao, M.: Radiative properties of coated black carbon aerosols impacted by their microphysics, Journal of Quantitative Spectroscopy & Radiative Transfer, 241, 106718, 10.1016/j.jqsrt.2019.106718, 2020.

6. Line 83: I think the authors mean aqueous processing

Thanks for pointing this out, and "water processing" has been changed to "aqueous processing".

Methods:

1. Lines 126-127: I think the authors mean "intensity of elements including carbon and heavier elements". As written, this would imply that they cannot detect carbon with their EDS.

Thanks for the comment. The sentence has been revised to "The EDS is coupled with TEM to detect the intensity of elements including carbon and heavier elements ($Z \geq 6$)."

2. Line 133: what are particles with rim?

Just like the particles in Figure 1e and 1f, there is a ring outside. That is the rim.

3. Line 140 and throughout: change "S-rich" to "S-rich particles"

It has been changed accordingly.

4. Line 148: please also cite [Moffet et al., 2008]

It has been cited.

5. Line 156: are particles focused by an aerodynamic lens or nozzle inlet?

An aerodynamic lens. The description has been revised to "Particles entering SPAMS were first focused into a beam of particles through an aerodynamic lens, and then their flight velocities were determined by two continuous diode Nd:YAG laser beams (532 nm)."

6. Line 156: the authors should clarify here that the sizing region measures the terminal velocity of the particles. These velocities are converted to vacuum aerodynamic diameter (not vacuum dynamic size) via calibration with polystyrene spheres of known size.

We agree with the comment. It has been revised as suggested. Please refer to the lines 158-161:
Particles entering the SPAMS were first focused into a beam of particles through an aerodynamic lens, and then their flight velocities were determined by two continuous diode Nd:YAG laser beams (532 nm). Polystyrene spheres of known size were used as a standard substance to calibrate the vacuum dynamic size ($d_{va}$) of particles.

7. Line 158: the authors should specify that ions are separated and analyzed using a dual polarity time-of-flight mass analyzer.

It has been supplemented accordingly.

8. Line 159: change "mass spectrometry" to "mass spectra"

It has been changed accordingly.

9. Lines 159-161: please cite [Bhave et al., 2002; Gross et al., 2000]

They have been cited.

10. Section 2.4: a lot of detail is in the SI. The authors need to at least mention that the particle analysis methods, calibration methods, and particle type characteristics can be found in the SI.

Thanks for pointing out this. Such information has been included in the main text: "A detailed description of particle analysis methods and particle type characteristics can be found in the supporting information." Please refer to lines 168-169.

Results:

1. Lines 192: does the high abundance of aged mixture particles in the RES indicate a lot of collision/coalescence of droplets with refractory material?

Thanks for the comment. It is hard to confirm from the data collected in the present study. These particles may also be activated to droplets since they are internally mixed with hygroscopic materials such as sulfate.

2. Line 203: is an incomplete sentence

Thanks for pointing out this. The sentence, "And the samples of cloud event #2 sampled at noon", has been changed to "And the samples of cloud event #2 were collected at noon." Please refer to line 212.

3. Lines 235-243: I suggest moving this text to section 3.4

It has been revised as suggested.

4. Lines 249-253: The authors cannot completely rule out that larger S-OM particles were more likely to be activated based on their size alone.

Thanks for the comment. Indeed, our data cannot completely rule out the activation of larger S-OM particles. Instead, we discussed the possibility that such core-shell mixing structure in the RES may also be explained by the primary activation of S-OM particles with larger sizes. As shown in the main text, the fraction of OM-containing particles increases from 33% in the INT to 60% in the RES. It is unlikely due to the favorable activation of S-OM or aged refractory mixture, since mixing with OM generally lower the hygroscopicity of inorganic-dominant particles (e.g., S-rich) (Brooks et al., 2004; Pierce et al., 2012). OM coating at the same site has been shown to inhibit the CCN activation of soot-containing particles (Zhang et al., 2017a). Besides, Evidence from the collocated SPAMS shows that the ratios of relative peak area between organics and sulfate are similar between the INT and particles before the cloud event, whereas they are higher in the RES (Table S2). This is corresponding to the production of oxidized organics during in-cloud processes (Zhang et al., 2017b), consistent with the TEM-EDS results. And thus, we suspect that it is most probably attributed to the in-cloud formation of OM on the surface of some S-rich particles, shifting the dominant particle type from S-rich to S-OM particles.

5. Line 258: Please clarify this sentence. It seems that the authors are implying that they do not have OM mixed with sulfate even though they just mentioned S-OM particles.

Thanks for pointing out this. The sentence has been changed to "Such a mixing structure is similar to those observed in the Arctic, background, or rural atmosphere (Hiranuma et al., 2013; Li et al., 2016; Yu et al., 2019), but is different from other findings in polluted areas, where OM-containing particles mainly existed in homogenous-like and thinly coated structures (Li et al., 2016).". Please refer to lines 257-260.

6. Lines 268-270: this sentence needs to be rephrased. I do not understand it.

Thanks for the comment, and the sentence has been revised to "Droplets are expected to dissolve more volatile organic compounds (Chakraborty et al., 2016) with higher O/C, and the release of these compounds during droplet evaporation would result in underestimation of O/C in the RES."

7. Lines 277-279: did the SPAMS see higher peak areas of m/z +43, which was found to be indicative of SOA and associated with high concentrations of $O_3$ (see [X Qin et al., 2012]).

Thanks for the comments. We did find a higher relative peak area of $m/z$ $43C_2H_3O^+$ in the RES than that in the INT during cloud event #2 with higher concentration of $O_3$. Such information has also been included in the discussion:
"Consistently, the relative peak area of $m/z$ $43C_2H_3O^+$ in the RES is higher than that in the INT during cloud event #2 (Figure S7), indicative of the favorable formation of oxidized organic compounds (Qin et al., 2012; Zhang et al., 2017b)."

Qin, X., Pratt, K. A., Shields, L. G., Toner, S. M., and Prather, K. A.: Seasonal comparisons of single-particle chemical mixing state in Riverside, CA, Atmospheric Environment, 59, 587-596, 10.1016/j.atmosenv.2012.05.032, 2012.

Zhang, G., Lin, Q., Peng, L., Yang, Y., Fu, Y., Bi, X., Li, M., Chen, D., Chen, J., Cai, Z., Wang, X., Peng, P., Sheng, G., and Zhou, Z.: Insight into the in-cloud formation of oxalate based on in situ measurement by single particle mass spectrometry, Atmospheric Chemistry and Physics, 17, 13891-13901, 10.5194/acp-17-13891-2017, 2017b.

Figures

1. I suggest adding an asterisk by elements that were considered background (e.g., Si and Cu).

Thanks for the comment. We add the asterisk on the background elements including C, Si, and Cu.

[Figure]

Figure 1. TEM images and EDS spectra of individual RES and INT particles with different particle types: (a) S-rich; (b) S-OM; (c) refractory; (d) aged soot; (e) aged mineral dust; (f) aged metal; (g) aged fly ash; (h) aged refractory mixture. Asterisk (*) represents the background element.

SI

1. Section 3: what software was used to import SPAMS spectra into Matlab? How many particles were analyzed?

The information on particle sizes and mass spectra is imported into the Matlab for subsequent analysis using the FATES toolkit (Sultana et al., 2017). A total of 117,436 particles from the SPAMS were analyzed. Such information has been included in section 3 of SI as suggested.

Sultana, C. M., Cornwell, G. C., Rodriguez, P., and Prather, K. A.: FATES: a flexible analysis toolkit for the exploration of single-particle mass spectrometer data, Atmos. Meas. Tech., 10, 1323-1334, doi:10.5194/amt-10-1323-2017, 2017.

2. Section 3 also needs more citations for the characterization of these particle types. I suggest citing [Denkenberger et al., 2007; X Qin et al., 2012; X Y Qin and Prather, 2006] for the EC and HMOC particle types. I suggest [Silva et al., 2000] for the dust particle type. I suggest [Gaston et al., 2011] for the sea salt particle type. I suggest [Angelino et al., 2001; Pratt et al., 2009] for the amine particle type.

We agree with the comments and have cited these references as suggested.

**Comments from Review #2**

Thank you for responding to the comments! I do think that there is still some confusion as the role of ionization efficiency and the complexities in equating concentration to relative intensity in a mass spectrum. I do think this should be clarified before the paper is published.

We agree with the comment. To make it clear, we have clarified the limitation of using relative peak area as a proxy for the relative abundance of the measured species in Section 2.4:

[revised manuscript text omitted]

**1 Air mass backward trajectories and meteorology conditions**

The backward trajectory and the height (above sea level) of air masses during sampling were calculated by the Hybrid Single Particle Lagrangian Integrated Trajectory (HYSPLIT) model (http://ready.arl.noaa.gov). During three cloud events, the sampling site was greatly influenced by air masses from Southeast Asia, northern China and the South China Sea. Compared with the cloud event #1, the air masses of cloud event #2 and #3 passed through a relatively low path on the way to the sampling site. Thus, the air masses of cloud event #2 and #3 were affected more by the ground anthropogenic emissions. The ambient temperature at the sampling station varied from 12.1 to 18.6 °C during three cloud events. All samples were collected during the stable period of cloud events, when the mass concentration of $PM_{2.5}$ was less than 5 $\mu$g m$^{-3}$ and visibility was less than 100 m. The concentrations of $PM_{2.5}$ during cloud event #1 were lower than those during cloud event #2 and #3. Consistently, the mean concentrations of $O_3$, $SO_2$ and $NO_X$ were higher in the cloud event #2 and #3 (Table S1).

**2 The size distribution of RES and INT**

In this study, a $PM_{2.5}$ cyclone inlet and a GCVI (ground-based counterflow virtual impactor) inlet were used to collect INT and RES, which is similar to Cozic et al. (2007). Additionally, the particle size in this study refers to as ECD (equivalent circle diameter) obtained from TEM images, which is larger than ESD (equivalent spherical diameter). Liu et al. (2018) showed that the ECD of individual dry particles on the substrate is 0.4952 times that of the ESD.

The size distribution data shows a higher median diameter of RES (1.20 $\mu$m) than INT (0.63 $\mu$m) (Figure S2), which are higher than those (0.8 and 0.45 $\mu$m, respectively) at Mount Tai in northern China (Li et al., 2011). This could be because Mount Tai is located in an industrial area, whereas our site represents a background region mainly influenced by long-range transport. Additionally, the formation of secondary compounds during cloud events increases the size of RES (Zhang et al., 2017).

The size distribution of different particle types revealed that S-rich and aged soot particles were predominant in smaller size segments, and aged mixture particles in larger size segments (Figure S8). Likewise, the size-resolved number fractions of different particle types from the results of the SPAMS also showed that the BC-containing particles were mainly distributed between 0.1 and 1.3 $\mu$m, representing ~80% of the submicron RES and ~73% of the submicron INT population, respectively (Figure S9).

**3 Identification of several types of particles within RES and INT measured by SPAMS**

The information on particle sizes and mass spectra is imported into the Matlab for subsequent analysis using the FATES toolkit (Sultana et al., 2017). A total of 117,436 particles from the SPAMS were analyzed. All the particles with bipolar mass spectra and the size range of $d_{va}$ 0.1–1.9 μm were classified several clusters by an adaptive resonance theory neural network (ART-2a) with a learning rate of 0.05, a vigilance factor of 0.8 and 20 iterations, and merged similar clusters manually. Ten characteristic particle types (Figure S1) were obtained, including BC (black carbon)-containing, OC (organic carbon), HMOC

(highly molecular organic carbon), Dust, K-rich, Metal, Na-K, Amines, SS (sea salt) and Others. BC- containing particles are characterized by elemental carbon cluster ions ($m/z$ 12C$^\pm$, 24C$_2^\pm$, 36C$_3^\pm$, 48C$_4^\pm$, …)

(Arndt et al., 2017). OC particles mainly contain fragment ions of organics ($m/z$ 27C$_2$H$_3^+$, 37C$_3$H$^+$,

43C$_2$H$_3$O$^+$, -26CN$^-$, …) (Denkenberger et al., 2007; Qin et al., 2012). The mass spectra of HMOC

particles show the presence of peaks of OC particles and some other organic peaks (such as $m/z$ 77C$_6$H$_5^+$,

91C$_7$H$_7^+$). Furthermore, HMOC particles are distinguished from OC particles by marked ion fragments detected in range of $m/z$ > 100 (Qin and Prather, 2006). Dust particles present significant ions at $m/z$

27Al$^+$, 40Ca$^+$ and 56CaO$^+$/Fe$^+$ (Silva et al., 2000). K-rich particles are identified according to the strong signal at $m/z$ 39K$^+$ only in positive mass spectra. Metal particles show the presence of metal ion peaks (such as Fe$^+$ ($m/z$ 54 and 56), Mn$^+$ ($m/z$ 55), Pb$^+$ ($m/z$ 206, 207 and 208)) in positive mass spectra. Na-K

particles are characterized by peaks at $m/z$ 23Na$^+$, 39K$^+$, and less intense peaks at $m/z$ -46NO$_2^-$, -62NO$_3^-$,

-97HSO$_4^-$. The mass spectra of amines particles contain ions signals at $m/z$ 59N(CH$_3$)$_3^+$, 86C$_5$H$_{12}$N$^+$,

101C$_6$H$_{15}$N$^+$ (Angelino et al., 2001; Pratt et al., 2009). SS particles are mainly composed of ions peaks at $m/z$ 23Na$^+$, 46Na$_2^+$, 62Na$_2$O$^+$, 63Na$_2$OH$^+$ and 81Na$_2$Cl$^+$ (Gaston et al., 2011). Most particles are observed to be internally mixed with sulfate and nitrate ($m/z$ -46, -62, -97). Particles with inconspicuous mass spectrum characteristics are named as others. Specific classification criteria were described in detail elsewhere (Zhang et al., 2015).

[Figure]

**Figure S1. Average positive and negative mass spectra of the main particle types (i.e., BC-containing, OC,**

**HMOC, Dust, K-rich, Metal, Na-K, Amines, SS) measured by SPAMS.**

[Figure]

**Figure S2. The chemical composition of RES measured by the SPAMS during cloud event #2 and #3.**

[Figure]

Figure S3. Time series of the chemical composition of RES and INT measured by the SPAMS during cloud events #2 and #3.

[Figure]

**Figure S4. Typical TEM images of soot particles in the RES (a-d) and INT (e-h).**

[Figure]

Figure S5. The size distribution of S-rich and S-OM particles. There are few S-rich particles with the size of less than 0.2 μm, and the median size are 0.56 μm and 0.76 μm for S-rich and S-OM particles, respectively.

[Figure]

Figure S6. Size distribution of RES and INT during cloud event #2 and #3. There are more INT particles when the size is less than 0.8 μm, and more RES particles when the size is larger than 0.8 μm.

[Figure]

**Figure S7. Average positive and negative mass spectra of OM particles (OC and HMOC) of RES and INT**

**particles measured by the SPAMS during cloud events #2 and #3.**

[Figure]

**Figure S8. Size-resolved number fraction distributions of RES and INT by TEM/EDS.**

[Figure]

Figure S9. Size-resolved number fraction distributions of RES and INT by the SPAMS.

**Table S1.** The concentration of $NO_X$, $SO_2$, $O_3$, $PM_{10}$ and $PM_{2.5}$ during three cloud events.

| cloud event | $NO_X$ (ppb) | $SO_2$ (ppb) | $O_3$ (ppb) | $PM_{10}$ ($\mu g\ m^{-3}$) | $PM_{2.5}$ ($\mu g\ m^{-3}$) |
|:---:|:---:|:---:|:---:|:---:|:---:|
| #1 | 2.6 | 0.4 | 30.5 | 3.6 | 1.1 |
| #2 | 3.5 | 1.2 | 39.1 | 4.8 | 1.9 |
| #3 | 4.3 | 0.6 | 34.4 | 11.4 | 4.7 |

**Table S2.** The ratios of relative peak area between organics (*m/z* 27, 29, 37, 43, 50, 51, 61, 63) and sulfate (*m/z* -97) of OM particles (OC and HMOC) during in-cloud (RES and INT) and pre-cloud (Ambient)

periods.

| | RES | INT | Ambient |
|---|---|---|---|
| Organics/Sulfates | 1.68 | 1.57 | 1.59 |

**Table S3.** Morphological descriptors of soot particles within RES and INT.

| parameters | $A_p$ | $d_p$ | $L_{max}$ | $N$ | $D_f$ | $k_g$ |
|:---:|:---:|:---:|:---:|:---:|:---:|:---:|
| RES | 1658(175) | 43(2) | 255(12) | 66(8) | 1.82(0.12) | 3.5(0.08) |
| INT | 1842(133) | 46(2) | 316(16) | 68(6) | 2.11(0.09) | 2.72(0.05) |

$A_p$, mean projected area of the monomer; $d_p$, monomer diameter; $L_{max}$, maximum length of soot aggregates; $N$, number of monomers in a soot aggregate; $D_f$, mass fractal dimension; $k_g$, structural coefficient. In parentheses are the standard error of $A_p$, $d_p$, $L_{max}$, $N$, $D_f$ and $k_g$.

**Table S4.** Overlap ($\delta$), constant ($k_a$) and empirical exponent ($\alpha$).

| parameters | $\delta$ | $k_a$ | $\alpha$ |
|:---:|:---:|:---:|:---:|
| RES | 1.54 | 1.52 | 1.13 |
| INT | 1.4 | 1.44 | 1.11 |